# MODEL-FREE REINFORCEMENT LEARNING THAT TRANSFERS USING RANDOM FEATURES

## ABSTRACT

Reinforcement learning (RL) algorithms have the potential not only for synthesizing complex control behaviors, but also for transfer across tasks. Typical model-free RL algorithms are usually good at solving individual problems with high dimensional state-spaces or long horizons, but can struggle to transfer across tasks with different reward functions. Model-based RL algorithms, on the other hand, naturally enable transfer across different reward functions, but struggle to scale to settings with long horizons and/or high dimensional observations. In this work, we propose a new way to transfer behaviors across tasks with different reward functions, displaying the benefits of model-free RL algorithms with the transferability of model-based RL. In particular, we show how a careful combination of model-free RL using randomly sampled features as reward is able to implicitly model long-horizon environment dynamics. Model-predictive control using these implicit models enables quick adaptation to problems with new reward functions, while scaling to problems with high dimensional observations and long horizons. Our method can be trained on offline datasets without reward labels, and quickly deployed on new tasks, making it more widely applicable than typical methods for both model-free *and* model-based RL. We validate that our proposed algorithm enables transfer across tasks in a variety of robotics and analytic domains.

## 1 INTRODUCTION

Reinforcement learning (RL) algorithms have been shown to successfully synthesize complex behavior in single-task sequential decision-making problems [1, 2, 3], but more importantly have the potential for broad generalization across problems. However, many RL algorithms are deployed as specialists — they solve single tasks and are not prepared for reusing their interactions. In this work, we specifically focus on the problem of transferring information across problems where the environment dynamics are shared, but the reward function is changing. This problem setting is reflective of a number of scenarios that may be encountered in real-world settings such as robotics. For instance, in tabletop robotic manipulation, different tasks like pulling an object, pushing an object, picking it up, and pushing to different locations, all share the same transition dynamics, but involve a changing reward function. We hence ask the question — can we reuse information across these tasks in a way that scales to high dimensional, longer horizon problems?

When considering how to tackle this problem, a natural possibility is to consider direct policy search [4, 5]. Typical policy search algorithms can achieve good performance for solving a single task, but entangle the dynamics and reward, in the sense that the policy one searches for is optimal for a particular reward but may be highly suboptimal in new scenarios. Other model-free RL algorithms like actor-critic methods [6, 7, 8] or Q-learning [9, 1] may exacerbate this issue, with learned Q-functions entangling dynamics, rewards, and policies. For new scenarios, an ideal algorithm should be able to disentangle and retain the elements of shared dynamics, while being able to easily substitute in new rewards.

A natural fit to disentangle dynamics and rewards are model-based RL algorithms [10, 11, 12, 13, 14]. These algorithms usually learn a single-step model of transition dynamics and leverage this learned model to perform planning [15, 12, 11, 16]. These models are naturally modular and can be used to *re-plan* behaviors for new rewards. However, one-step dynamics models are brittle and suffer from challenges in compounding error [17, 18].

In this work, we ask — can we build reinforcement learning algorithms that disentangle dynamics, rewards, and policies for transfer across problems but retain the ability to solve problems with high dimensional observations and long horizons? In particular, we propose an algorithm that can train on large offline datasets of transitions in an environment at training time to implicit model transition dynamics, and then quickly perform decision making on a variety of different new tasks with varying reward functions that may be encountered at test time.

Specifically, we propose to model the long-term behavior of randomly chosen basis functions (often called cumulants) of the environment state and action, under open-loop control, using what we term Q-basis functions. These Q-basis functions can be easily recombined to infer the true Q function for tasks with arbitrary rewards by simply solving a *linear regression* problem. Intuitively, this suggests that rather than predicting the evolution of the entire state step by step, predicting the accumulated long-term future of many random features of the state contains information equivalent to a dynamics model, thereby forming an "implicit model" that can transfer. These implicit models scale better with horizon and environment dimensionality than typical one-step dynamics models, while retaining the benefits of transferability and modularity.

Our proposed algorithm *Random Features for Model-Free Planning* (RaMP) allows us to leverage an unlabelled offline dataset to learn reward-agnostic implicit models that can quickly solve new tasks involving different reward functions in the same shared environment dynamics. We show the efficacy of this method on a number of tasks for robotic manipulation and locomotion in simulation, and highlight how RaMP provides a more general paradigm than typical generalizations of model-based or model-free reinforcement learning.

## 1.1 RELATED WORK

Model-based RL is naturally suited for this transfer learning setting, by explicitly learning a model of the transition dynamics and the reward function [12, 15, 11, 19, 16, 20, 21]. These models are typically learned via supervised learning on one-step transitions and are then used to extract control actions via planning [22, 23] or trajectory optimization [24, 25, 26]. The key challenge in scaling lies in the fact that they sequentially feed model predictions back into the model for sampling [27, 18, 17]. This can often lead to compounding errors [17, 18, 28], which grows with the horizon length unfavorably. In contrast, our work does not require autoregressive sampling, but directly models long term behavior, and is easier to scale to longer horizons and higher dimensions.

On the other hand, model-free RL often avoids the challenge of compounding error by directly modeling either policies or Q-values [4, 29, 5, 30, 1, 7] and more easily scales to higher dimensional state spaces [1, 31, 5]. However, this entangles rewards, dynamics, and policies, making it challenging to directly use for transfer. While certain attempts have been made at building model-free methods that generalize across rewards, such as goal-conditioned value functions [32, 33, 34, 35] or multi-task policies [36, 37], they only apply to restricted classes of reward functions and particular training distributions. Our work aims to obtain the best of both worlds (model-based and model-free RL), learning a disentangled representation of dynamics that is independent of rewards and policies, but using a model-free algorithm for learning.

Our notion of long-term dynamics is connected to the notion of state-action occupancy measure [38, 39], often used for off-policy evaluation and importance sampling methods in RL. These methods often try to directly estimate either densities or density ratios [14, 38, 39]. Our work simply learns the long-term accumulation of random features, without requiring any notion of normalized densities.

Perhaps most closely related work to ours is the framework of *successor features*, that considers transfer from a fixed set of source tasks to new target tasks [40, 41, 42, 43]. Like our work, the successor features framework leverages linearity of rewards to disentangle long-term dynamics from rewards using model-free RL. However, transfer using successor features is critically dependent on choosing (or learning) the right featurization and entangles the policy. Our work leverages random features and open-loop policies to allow for transfer across arbitrary policies and rewards.

## 2 BACKGROUND AND SETUP

**Formalism:** We consider the standard Markov decision process (MDP) as characterized by a tuple $\mathcal{M} = (\mathcal{S}, \mathcal{A}, \mathcal{T}, R, \gamma, \mu)$, with state space $\mathcal{S}$, action space $\mathcal{A}$, transition dynamics $\mathcal{T} : \mathcal{S} \times \mathcal{A} \rightarrow$

$\Delta(\mathcal{S})$, reward function $\mathcal{R} : \mathcal{S} \times \mathcal{A} \rightarrow \Delta([-R_{\max}, R_{\max}])$, discount factor $\gamma \in [0, 1)$, and initial state distribution $\mu \in \Delta(\mathcal{S})$. The goal is to learn a policy $\pi : \mathcal{S} \rightarrow \Delta(\mathcal{A})$, such that it maximizes the expected discounted accumulated rewards, i.e., solves $\max_\pi \mathbb{E}_\pi \left[ \sum_{h=1}^\infty \gamma^{h-1} r_h \right]$ with $r_h :=$ $r(s_h, a_h) \sim R_{s_h,a_h} = \Pr(\cdot \mid s_h, a_h)$. Hereafter, we will refer to an MDP and a *task* interchangeably.

**Estimating Q-functions:** Given an MDP $\mathcal{M}$, one can define the state-action $Q$-value function under any policy $\pi$ as $Q_\pi(s, a) := \mathbb{E}_{\substack{a_h \sim \pi(\cdot \mid s_h) \\ s_{h+1} \sim \mathcal{T}(\cdot \mid s_h, a_h)}} \left[ \sum_{h=1}^\infty \gamma^{h-1} r_h \mid s_1 = s, a_1 = a \right]$ which denotes the expected accumulated reward under policy $\pi$, when starting from state-action pair $(s, a)$. Similarly, one can also define the *multi-step ($\tau$-step) Q-function* $Q_\pi(s, \tilde{a}_1, \tilde{a}_2, \cdots, \tilde{a}_\tau) = \mathbb{E}_{\substack{a_{\tau+h} \sim \pi(\cdot \mid s_{\tau+h}) \\ s_{h+1} \sim \mathcal{T}(\cdot \mid s_h, a_h)}} \left[ \sum_{h=1}^\infty \gamma^{h-1} r_h \mid s_1 = s, a_1 = \tilde{a}_1, a_2 = \tilde{a}_2, \cdots, a_\tau = \tilde{a}_\tau \right]$.

One can estimate the $Q_\pi$ by Monte-Carlo sampling of the trajectories under $\pi$, i.e., by solving

$$\min_{\widehat{Q} \in \mathcal{Q}} \; \frac{1}{N} \sum_{j=1}^N \left\| \widehat{Q}(s, \tilde{a}_1^j, \tilde{a}_2^j, \cdots, \tilde{a}_\tau^j) - \frac{1}{M} \sum_{m=1}^M \sum_{h=1}^\infty \gamma^{h-1} r_h^{m,j} \right\|_2^2, \tag{2.1}$$

where $\mathcal{Q}$ is some function class for $Q$-value estimation, which in practice is some parametric function class, e.g., neural networks; $r_h^{m,j} \sim R_{s_h^{m,j}, a_h^{m,j}}$ and $(s_h^{m,j}, a_h^{m,j})$ come from $MN$ trajectories that are generated by $N$ open-loop action sequences $\{(\tilde{a}_1^j, \tilde{a}_2^j, \cdots, \tilde{a}_\tau^j)\}_{j=1}^N$. For each sequence there are $M$ trajectories starting from it, and following policy $\pi$ onwards, to estimate the $\tau$-step Q-function. A large body of work considers finding this $Q$-function using dynamic programming, but for the sake of simplicity, this work will only consider Monte-Carlo estimation.

In practice, the infinite-horizon estimator in (2.1) can be hard to obtain. We hence use a finite-horizon approximation of $Q_\pi$ (of length $H$), denoted by $Q_\pi^H$, in learning. Note that if one chooses $H = \tau$, then the $\tau$-step Q-function defined above becomes $Q_\pi^H(s, \tilde{a}_1, \tilde{a}_2, \cdots, \tilde{a}_H) :=$ $\mathbb{E}_{s_{h+1} \sim \mathcal{T}(\cdot \mid s_h, a_h)} \left[ \sum_{h=1}^H \gamma^{h-1} r_h \mid s_1 = s, a_1 = \tilde{a}_1, \cdots, a_H = \tilde{a}_H \right]$. Note that in this case, the $Q$-function is irrelevant of the policy $\pi$, denoted by $Q^H$, and is just the expected accumulated reward under the open-loop action sequence $(\tilde{a}_1, \tilde{a}_2, \cdots, \tilde{a}_H)$. This Q-function can be used to score how "good" a sequence of actions will be, which in turn can be used for planning.

## 2.1 Problem Setup

We consider a transfer and offline RL scenario, where we assume access to an *offline* dataset consisting of several episodes $\mathcal{D} = \{(s_h^m, a_h^m, s_{h+1}^m)\}_{h \in [H], m \in [M]}$. This dataset assumes that all transitions are collected under the same transition dynamics $\mathcal{T}$, but otherwise does not require labels for rewards, and may come from multiple different behavior policies as well. Here $H$ is the length of the trajectories, which is large enough, e.g., of order $\mathcal{O}(1/(1-\gamma))$ to approximate the infinite-horizon setting; $M$ is the total number of trajectories.

The goal is to make the best use of the dataset $\mathcal{D}$, and generalize the learned experience to improve the performance on a new task $\mathcal{M}$, with the same transition dynamics $\mathcal{T}$ but arbitrary reward functions $\mathcal{R}$. Note that unlike some related work [40, 44], we make *no* assumption on the reward functions of the MDPs that generate $\mathcal{D}$, i.e., these MDPs do not have to share any *structure* of the reward functions, e.g., being linear in some common features. In fact, the samples of the rewards that correspond to the trajectories in $\mathcal{D}$ are not even necessary. The goal of the learning problem is to *pre-train* on the offline dataset such that we can enable very quick (even zero-shot) adaptation to the new reward functions encountered at test time.

## 3 RaMP: Learning Implicit Models for Cross-Reward Transfer with Model-Free Techniques

In this section, we introduce our algorithm, Randomized features for Model-free Planning (RaMP), to solve the problem described in §2 – learning a model of long-term dynamics that enables transfer to tasks labeled with arbitrary new rewards, while mitigating challenges with compounding error.

Figure 1: `RaMP`: Depiction of our proposed method for transferring behavior across tasks by leveraging model-free learning of random features. At training time, Q-basis functions are trained on accumulated random features. At test time, adaptation is performed by solving linear regression and recombining basis functions, followed by online planning with MPC.

We start by arguing where model-based and model-free algorithms fall short. Model-based RL approaches estimate the transition dynamics $\mathcal{T}$ using the data in $\mathcal{D}$, and plan in the estimated model. The key advantage of this approach is that it is *reward-agnostic*, and has the potential to easily generalize to multiple tasks. Unfortunately, since the model outputs are fed back into the model for multi-step planning, it is subject to compounding error of one-step dynamics models [17].

In contrast, one can resort to *model-free RL* approaches, e.g., Q-learning or policy optimization methods [6, 9, 1, 5, 4], to directly optimize the value of interest. These methods are less subject to the challenges of compounding error than most model-based ones. Empirically, learning neural networks to predict the Q-function (a *scalar* for each $(s, a)$), can be much easier than to predict the next state (which can be a *high-dimensional* vector, e.g., image). However, these methods cannot be used directly to transfer across different tasks with different rewards, as they are designed to be *reward-dependent*. This raises the natural question: *Is there a model-free approach that can mitigate the challenges of compounding error and can transfer across tasks painlessly?*

The key insight we advocate is that if instead of modeling long-term accumulation of some specific reward as a Q-function, we directly model long-term accumulation of many random features of state-actions under arbitrary open-loop action sequences. This can effectively disentangle transition dynamics, reward, and policies being evaluated, and potentially allow for transfer across tasks. Each long-term accumulation of random features is referred to as an element of a "random" Q-basis, and can be learned with simple modifications to typical model-free RL algorithms.

At *training time*, the offline dataset $\mathcal{D}$ can be used to learn a set of "random" Q-basis functions for different random features. This effectively forms an "implicit model", as it carries information about how the dynamics propagates, without being tied to any particular reward function or policy. At *test time*, given a new reward function, we can recombine Q-basis functions linearly to effectively approximate the true reward-specific Q-function. This inferred Q-function can then be used for planning for the new task.

### 3.1 OFFLINE TRAINING: LEARNING RANDOM Q-FUNCTIONS FROM UNLABELLED DATA

Given a dataset of transitions without reward labels, the goal of this phase is to model the long-term accumulation of random features under random state-action sequences. With no prior knowledge about the downstream test-time rewards, the random features being modeled must be expressive and universal in their coverage, so that any possible test-time rewards can be reconstructed from these random features by linear regression. As suggested in [45, 46, 47], random features can be powerful in representing nonlinear functions, i.e., any test-time reward function in our case, as their linear combinations. In particular, suppose we have $K$ neural networks $\phi(\cdot, \cdot; \theta_k) : \mathcal{S} \times \mathcal{A} \to \mathbb{R}$ with weights $\theta_k \in \mathbb{R}^d$ and $k \in [K]$, where $\theta_k$ are *randomly* i.i.d. sampled from some distribution $p$. Sampling $K$ such weights $\theta_k$ with $k \in [K]$ yields a vector of scalar functions $[\phi(\cdot, \cdot; \theta_k)]_{k \in [K]} \in \mathbb{R}^K$ for any $(s, a)$, which can be used as random features whose accumulation through dynamics can be used to model Q-basis functions.

To model the long-term accumulation of each of these random features, we note that they can be treated as reward functions in model-free RL, and the machinery of Q-functions can be reused to

learn their long-term accumulation. As discussed in [48], model-free RL algorithms can be used to model the evolution of arbitrary functions (called "cumulants") of the state. Therefore, we can learn a set of $K$ Q-basis functions, with each of them corresponding to a particular random feature.

We note that this definition of a Q-basis function is tied to a particular policy $\pi$ that generates the trajectory. To transfer, one needs to predict the accumulated random features under new sequences of actions, as the optimal policy for the new task is likely to not be within the span of policies seen in training. To allow the modeling of cumulants that is independent of particular policies, we propose to learn *open-loop* Q-basis functions for each of the random features (as discussed in Section §2), which is policy-agnostic, and can be used to search for optimal actions in new tasks.

To actually learn these Q-basis functions (one for each random feature), we opt to use Monte-Carlo methods for simplicity. We generate a new dataset $\mathcal{D}_\phi$ from $\mathcal{D}$, with $\mathcal{D}_\phi = \{((s_1^m, a_{1:H}^m), \sum_{h \in [H]} \gamma^{h-1} \phi(s_h^m, a_h^m; \theta_k))\}_{m \in [M], k \in [K]}$. Here we use $\sum_{h \in [H]} \gamma^{h-1} \phi(s_h^m, a_h^m; \theta_k)$ as the accumulated cumulants for open-loop action sequences $\{a_1, \cdots, a_H\}$ taken from state $s_1$. We then use $K$ function approximators representing each of the $K$ Q-basis functions, e.g., neural networks $\psi(\cdot, \cdot; \nu_k) : \mathcal{S} \times \mathcal{A}^H \to \mathbb{R}$ for $k \in [K]$, to fit the accumulated cumulants. Specifically, we minimize the following loss

$$\min_{\{\nu_k\}_{k \in [K]}} \frac{1}{M} \sum_{m \in [M], k \in [K]} \left( \psi(s_1^m, a_{1:H}^m; \nu_k) - \sum_{h \in [H]} \gamma^{h-1} \phi(s_h^m, a_h^m; \theta_k) \right)^2. \quad (3.1)$$

These Q-basis functions can be recombined to approximate the Q-functions for true rewards at test time. The two key design decisions here are – (1) predicting the evolution of random features, rather than one-step modeling of state, and (2) predicting the accumulated random features under open-loop action sequences, rather than a closed-loop policy.

## 3.2 ONLINE PLANNING: INFERRING Q-FUNCTIONS WITH LINEAR REGRESSION AND PLANNING WITH MODEL-PREDICTIVE CONTROL

The goal of our learned Q-basis functions is to enable transfer to new tasks with arbitrary rewards. Any reward function can be approximately expressed as a linear combination of a sufficiently expressive and expansive set of random features. Given this linear approximation, we can recover an approximation to the Q-function for the true test-time reward by recombining the random Q-basis functions linearly. Therefore, we can obtain the test-time Q-function by solving a *simple* linear regression problem. This inferred Q-function can then be used to obtain an optimal sequence of actions through planning.

### 3.2.1 REWARD FUNCTION FITTING WITH RANDOMIZED FEATURES

We first learn how to express the reward function for the new task as a linear combination of the random features. This can be done by solving a linear regression problem to find the coefficient vector $w = [w_1, \cdots, w_K]^\top$ that approximates the new task's reward function as a linear combination of the random features. Specifically, we minimize the following loss

$$w^* = \underset{w}{\operatorname{argmin}} \ \frac{1}{MH} \sum_{h \in [H], m \in [M]} \left( r(s_h^m, a_h^m) - \sum_{k \in [K]} w_k \phi(s_h^m, a_h^m; \theta_k) \right)^2 + \lambda \|w\|_2^2, \quad (3.2)$$

where $\lambda \geq 0$ is the regularization coefficient, and $r(s_h^m, a_h^m) \sim R_{s_h^m, a_h^m}$. Due to the use of random features, Eq. (3.2) is a *ridge regression* problem, and can be solved efficiently. In case the reward labels are being obtained on the fly during online data collection, we can leverage an online least squares algorithm [49] to continually improve our estimate of $w$ without re-computing regression result from scratch as the number of samples grows.

Given these weights, it is easy to estimate an approximate open-loop Q-function for the true reward on the new task by linearly combining the Q-basis functions learned in the offline training phase $\{\psi(\cdot, \cdot; \nu_k^*)\}_{k \in [K]}$ according to the *same* coefficient vector $w^*$. This follows from the additive nature of reward and linearity of expectation. In particular, if the reward function $r(s, a) = \sum_{k \in [K]} w_k^* \phi(s, a; \theta_k)$ holds approximately, which will be the case for large enough $K$ and rich enough $\phi$, then the approximate Q-function for the true test-time reward under

the sequence $\{a_1, \cdots, a_H\}$ satisfies $Q^H(s_1, a_{1:H}) := \mathbb{E}_{s_{h+1} \sim \mathcal{T}(s_h, a_h)} \left[ \sum_{h \in [H]} \gamma^{h-1} R_{s_h, a_h} \right] \approx$ $\sum_{k \in [K]} w_k^* \psi(s_1, a_{1:H}; \nu_k^*)$, where $\{w_k^*\}_{k \in [K]}$ is the solution to the regression problem (Eqn (3.2)) and $\{\nu_k^*\}_{k \in [K]}$ is the solution to the Q-basis fitting problem (Eqn (3.1)).

### 3.2.2 PLANNING WITH MODEL-PREDICTIVE CONTROL

To obtain the optimal sequence of actions we can use the inferred approximate Q-function for the true reward $Q^H(s_1, a_{1:H})$ for online planning at each time $t$ in the new task: at state $s_t$, we conduct standard model-predictive control with random shooting, i.e., randomly generating $N$ sequences of actions $\{a_1^n, \cdots, a_H^n\}_{n \in [N]}$, and find the action sequence with the maximum Q-value such that

$$n_t^* \in \underset{n \in [N]}{\arg\max} \ \sum_{k \in [K]} w_k^* \psi(s_t, a_{t:t+H-1}^n; \nu_k^*). \tag{3.3}$$

We then execute $a_t^{n_t^*}$ from the sequence $n_t^*$, observe the new state $s_{t+1}$, and replan. Our algorithm is summarized in Algorithm 1. We refer readers to Appendix D for a detailed connection of our proposed method to existing work and Appendix A for detailed pseudocode.

### 3.3 THEORETICAL JUSTIFICATIONS

We now provide some theoretical justifications for the methodology we adopt. To avoid unnecessary nomenclature of measures and norms in infinite dimensions, we in this section consider the case that $\mathcal{S}$ and $\mathcal{A}$ are discrete (but can be enormously large). Due to space limitation, we present an abridged version of the results below, and defer the detailed versions and proofs in §C. We first state the following result on the expressiveness of random cumulants.

**Theorem 3.1** (Q-function approximation; Informal). Under standard coverage and sampling assumptions of offline dataset $\mathcal{D}$, and standard assumptions on the boundedness and continuity of random features $\phi(s, a; \theta)$, it follows that with horizon length $H = \widetilde{\Theta}(\frac{\log(R_{\max}/\epsilon)}{1-\gamma})$ and $M = \widetilde{\Theta}(\frac{1}{(1-\gamma)^3 \epsilon^4})$ episodes in dataset $\mathcal{D}$, and with $K = \widetilde{\Omega}((1-\gamma)^{-2} \epsilon^{-2})$ random features, we have that for any given reward function $R$, and any policy $\pi$

$$\|\widehat{Q}_\pi^H(w^*) - Q_\pi\|_\infty \leq \mathcal{O}(\epsilon) + \mathcal{O}\left(\sqrt{\inf_{f \in \mathcal{H}} \mathcal{E}(f)}\right)$$

with high probability, where for each $(s, a)$, $\widehat{Q}_\pi^H(s, a; w^*)$ is defined as $\widehat{Q}_\pi^H(s, a; w^*) := \mathbb{E}\left[ \sum_{h=1}^{H} \gamma^{h-1} \sum_{k \in [K]} w_k^* \phi(s_h, a_h; \theta_k) \, \Big| \, s_1 = s, a_1 = a \right]$, and can be estimated from the offline dataset $\mathcal{D}$; $\inf_{f \in \mathcal{H}} \mathcal{E}(f)$ is the infimum expected risk over the function class $\mathcal{H}$ induced by $\phi$.

Theorem 3.1 is an informal statement of the results in §C.2, which specifies the number of random features, the horizon length per episode, and the number of episodes, in order to approximate $Q_\pi$ accurately by using the data in $\mathcal{D}$, under any given reward function $R$ and policy $\pi$. Note that the number of random features is not excessive and is polynomial in problem parameters. We also note that the results can be improved under stronger assumptions of the sampling distributions $p$ and kernel function classes [47, 50].

Next, we justify the use of open-loop Q-functions in planning in a deterministic transition environment, which contains all the environments our empirical results we will evaluate later. Recall that for any given reward $R$, let $Q_\pi$ denote the Q-function under policy $\pi$. Note that with a slight abuse of notation, $Q_\pi$ can also be the multi-step Q-function (see definition in §2), and the meaning should be clear from the input, i.e., whether it is $Q_\pi(s, a)$ or $Q_\pi(s, a_1, \cdots, a_H)$.

**Theorem 3.2.** Let $\Pi$ be some subclass of Markov stationary policies, i.e., for any $\pi \in \Pi$, $\pi : \mathcal{S} \to \Delta(\mathcal{A})$. Suppose the transition dynamics $\mathcal{T}$ is deterministic. For any given reward $R$, denote the $H$-step policy obtained from $H$-step open-loop policy improvement over $\Pi$ as $\pi_H' : \mathcal{S} \to \mathcal{A}^H$, defined as $\pi_H'(s) \in \arg\max_{(a_1, \cdots, a_H) \in \mathcal{A}^H} \max_{\pi \in \Pi} Q_\pi(s, a_1, \cdots, a_H)$, for all $s \in \mathcal{S}$. Let $V_{\pi_H'}$ denote the

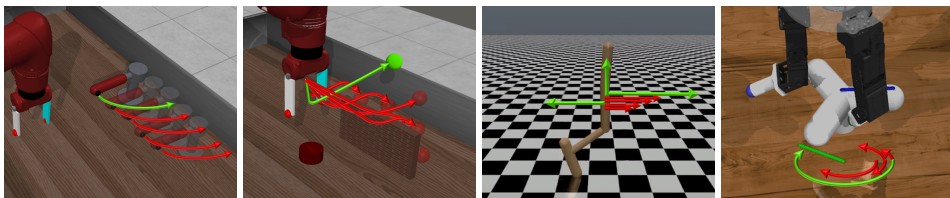

Figure 2: We evaluate our method on manipulation, locomotion, high dimensional action space environments. The green arrow in each environment indicates the online objective for policy transfer while the red arrows are offline objectives used to label rewards for the privileged dataset.

value function under the open-loop policy $\pi'_H$ (see formal definition in §C.1). Then, we have that for all $s \in \mathcal{S}$

$$V_{\pi'_H}(s) \geq \max_{a_{1:H}} \max_{\pi \in \Pi} Q_\pi(s, a_1, \cdots, a_H) \geq \max_a \max_{\pi \in \Pi} Q_\pi(s, a).$$

The proof of Theorem 3.2 can be found in §C.2. The result can be viewed as a generalization of the generalized policy iteration result in [40] to multi-step open-loop policies. Taking $\Pi$ to be the set of policies that generate the data, the result shows that the value function of the greedy open-loop policy improves over all the possible $H$-step open-loop policies, with the policy after step $H$ to be any policy in $\Pi$. Moreover, the value function by $\pi'_H$ also improves overall one-step policies if the policy after the first step onwards follows any policy in $\Pi$. This is due to the fact that $\Pi$ (coming from data) might be a much more restricted policy class than any open-loop sequence $a_{1:H}$.

# 4 EXPERIMENTAL EVALUATION

In this section, we aim to answer the following research questions: **(1)** Does `RaMP` allow for effective transfer of behaviors across tasks with varying rewards but shared dynamics?, **(2)** Does `RaMP` scale to domains with high dimensional observation spaces and longer horizons?, **(3)** Does `RaMP` scale to domains with high dimensional action space? **(4)** Which design decisions in `RaMP` enable better transfer and scaling?

## 4.1 EXPERIMENT SETUP

Across several domains, we evaluate the ability of `RaMP` to leverage the knowledge of shared dynamics from an offline dataset to quickly solve new tasks with arbitrary rewards.

**Offline Dataset Construction:** For each domain, we have an offline dataset collected by a behavior policy as described in Appendix B.2. Typically this behavior policy is a mixture of noisy policies accomplishing different objectives in each domain. Although `RaMP` and other model-based methods do not require knowledge of any reward from the offline dataset and simply require transitions, other baseline comparisons will require privileged information. Baseline comparison methods like model-free RL and successor features require the provision of a set of training objectives, as well as rewards labeled for these objectives on state-actions from the offline dataset. We call such objectives 'offline objectives' and a dataset annotated with these offline objectives and rewards a privileged dataset.

**Test-time adaptation:** At test-time, we select a novel reward function for online adaptation, referred to as 'online objective' below. The online objective may correspond to rewards conditioned on different goals or even arbitrary rewards, depending on the domain. Importantly, the online objective need not be drawn from the same distribution as the privileged offline objectives above.

Given this problem setup, we compare `RaMP` with a variety of baselines. (1) MBPO [13] is a model-based reinforcement learning method that learns a standard one-step dynamics model and uses actor-critic methods to plan in the models. We pre-train the dynamics model for MBPO on the offline dataset before running the full algorithm on the testing environment. (2) Successor feature (SF) [40] is a framework for transfer learning in RL as described in Sec. 1.1. SF typically assumes access to a set of policies towards different goals along with a learned featurization, so we provide it with the privileged dataset to learn a set of policies corresponding to the offline objectives using offline reinforcement learning [51]. We also learn successor features with the privileged dataset [40]. (3) CQL [51]: As an oracle comparison, we compare with a goal-conditioned variant of an offline RL algorithm (CQL). CQL is a model-free offline RL algorithm that learns policies from offline data.

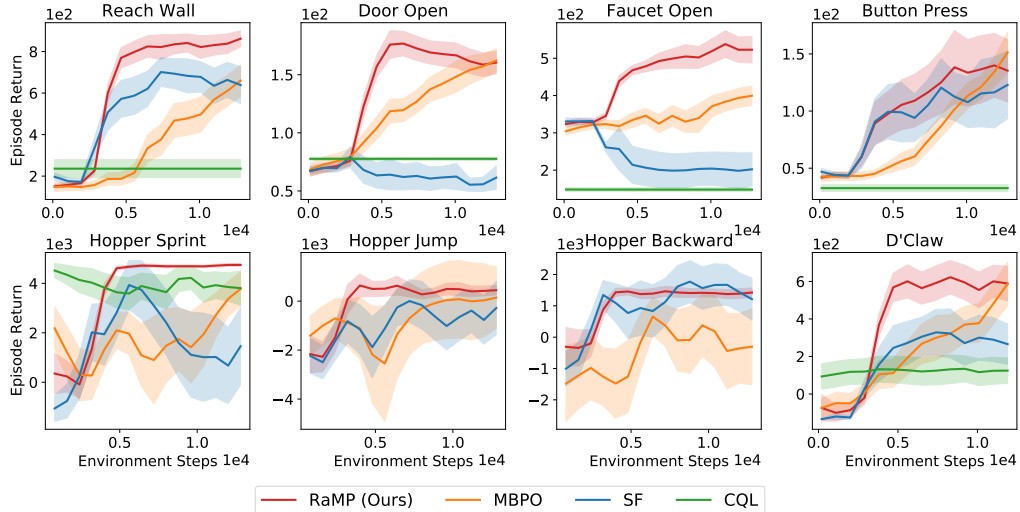

Figure 3: Reward transfer results on Metaworld, Hopper and D'Claw environments. `RaMP` adapts to novel rewards more rapidly than MBPO, Successor Features, and CQL baselines. More experiments are in appendix.

While model-free offline RL naturally struggles to adapt to arbitrarily changing rewards, we instead afford CQL additional privileges by providing it with information about the goal at both training and testing time. CQL is then trained on the distribution of training goals on the offline dataset, and finetuned on the new goal provided at test time. In this sense, the CQL comparison is assuming access to more information than `RaMP`. Each method is benchmarked on each domain with 9 seeds.

## 4.2 TRANSFER TO NOVEL REWARDS

We first evaluate the ability of `RaMP` to learn from an offline dataset and quickly adapt to novel test rewards in 4 robotic manipulation environments from meta-world [52]. We consider skills like reaching a target across the wall, opening a door, turning on a faucet, and pressing a button while avoiding obstacles, which are challenging for typical model-based RL algorithms (Fig. 2).

Each domain features 50 different possible goal configurations, each associated with a different reward but the same dynamics. The privileged offline objectives consist of 25 goal configurations as described in Sec.4.1. The test-time reward functions are drawn from the remaining 25 "out-of-distribution" reward functions. We refer the reader to Appendix B.1 for details of this setup.

As shown in Fig 3, our method adapts to test reward most quickly across all four domains. MBPO slowly catches up with our performance with more samples, since it still needs to learn the Q function from scratch even with the dynamics branch trained. In multiple environments, successor features barely transfer to the online objective as it entangles policies that aren't close to those needed for the online objective. Goal-conditioned CQL performs poorly in all tasks as it faces a hard time generalizing to out-of-distribution goals. In comparison, `RaMP` is able to deal with arbitrary sets of test time rewards, since it does not depend on the reward distribution at training time.

## 4.3 SCALING TO TASKS WITH LONGER HORIZONS

We further evaluate the ability of our method to scale to tasks with longer horizons. We consider locomotion domains such as the Hopper environment from OpenAI Gym [53]. We chose the offline objectives to be running forward at different velocities. The online objectives for adaptation correspond to novel skills such as standing, sprinting, jumping, or running backward. Among them, standing and sprinting are goal-conditioned objectives that correspond to running forward at zero and maximum speed, while jumping and running backward have drastically different objectives that are difficult to express as parametric "goals". Therefore, goal-conditioned methods like CQL are not applicable on jumping and running backward. As shown in Fig. 3, our method maintains the highest performance when adapting to drastically different online objectives, as it is designed to make no assumption about reward in the offline dataset, while avoiding compounding errors by directly modeling accumulated random features. MBPO fails to match the performance of `RaMP` since higher dimensional observation and longer horizon increase the compounding error of model-based

methods. We note that SF is performing reasonably well, likely because the method also reduces the compounding error compared to MBPO. Furthermore, its featurization is trained with privileged data and thus still captures useful information for the online objectives. In Appendix B.4, we further test our method on environments with even higher dimensional observations, such as image observations. We refer the reader to Appendix B.1 for further details.

## 4.4 Scaling to High Dimensional State-Action Spaces

To understand whether `RaMP` can scale to higher dimensional state-action spaces, we consider a dexterous manipulation domain (referred to as the D'Claw domain in Fig 2). This domain has a 9 DoF action space controlling each of the joints of the hand as well as a 16-dimensional state space including object position. The offline dataset is collected moving the object to different orientations, and the test-time rewards are tasked with moving the object to new orientations (as described in Appendix B.1). Fig 3 shows that both Q-estimation and planning with model-predictive control remain effective when action space is large.

## 4.5 Ablation of Design Choices

To understand what design decisions in `RaMP` enable better transfer and scaling, we conduct ablation studies on various domains, including an analytical 2D point goal-reaching environment and its variants (described in Appendix B.1), as well as the classic InvertedDoublePendulum domain and meta-world reaching. We report an extensive set of ablations in Appendix B.5.

|      | Hopper | Pendulum |
|------|--------|----------|
| Ours | **25.7 ± 5.4** | **65.1 ± 0.4** |
| MBRL | 50.2 ± 9.7 | 395.4 ± 5.8 |

Table 1: Policy evaluation error. Feedforward dynamics model suffers from compounding error that is particularly noticeable in domains with high action dimensions or chaotic dynamics. Our method achieves low approximation error in both domains.

|            | Point | Reach |
|------------|-------|-------|
| Random     | **34.0 ± 1.0** | **820.8 ± 142.0** |
| Gaussian   | -3.6 ± 7.4 | 188.6 ± 49.2 |
| Polynomial | 24.5 ± 5.8 | 162.9 ± 36.4 |

Table 2: Return for different features. Random features are able to approximate the true reward well across domains. Polynomial features work in simple environments but do not scale to complex rewards. Gaussian features are unable to express the reward.

**Reduction of compounding error with open-loop Q functions** We hypothesize that our method does not suffer from compounding errors in the same way that feedforward dynamics models do. In Table 1, we compare the approximation error of truncated Q values computed with (1) open-loop Q functions obtained as a linear combination of random cumulants (Ours), and (2) rollouts of a feedforward dynamics model (MBRL). We train the methods on offline data and evaluate on data from a novel task at test time. Note that this setting is analogous to performing policy evaluation on the behavioral policy induced by an offline dataset. We perform the comparison on one environment with high action dimension (Hopper) and one with chaotic dynamics (Pendulum). As shown in Table 1, our method outperforms feedforward dynamics models.

**Effect of different types of featurization** We experiment with three choices of projections: random features parametrized by a deep neural network, random features parametrized by a gaussian matrix, and polynomial features of state and action up to the second order. We evaluate these choices on Point and Metaworld Reach in Table 2. We see that NN-parametrized random features approximate the true reward well as a linear combination of random features for all three tasks. Polynomial features perform well on environments with simple rewards that are linear in polynomial basis, but struggle as the reward becomes more complex, while Gaussian features are rarely expressive enough.

## 5 Discussion

In this work, we introduce `RaMP`, a method for leveraging diverse prior offline data to learn models of long horizon dynamics behavior, while being able to naturally transfer across tasks with different reward functions. To do so, we combine the best elements of model-based and model-free reinforcement learning. By learning the long-term evolution of random features under open loop policies, we are able to disentangle dynamics, rewards, and policies. We show how this technique allows us to learn behavior that naturally transfers across tasks, even under misspecification of reward functions. Across a number of simulated robotics and control domains, `RaMP` achieves superior transfer

ability than baseline comparisons. In future work, we hope to explore how to combine `RaMP` with more powerful planning methods like [54] and dynamic program techniques for learning Q-basis functions [6].

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

# Supplementary Materials for
# "Model-free Reinforcement Learning that Transfers using Random Features"

## A  ALGORITHM PSEUDOCODE

---

**Algorithm 1** Model-free Transfer with Randomized Cumulants and Model-predictive Control

---

1: **Input**: Offline dataset $\mathcal{D}$ given by (2.1), distribution $p$ over $\mathbb{R}^d$, number of random features $K$

2: **Offline Training Phase:**

3: Randomly sample $\{\theta_k\}_{k \in [K]}$ with $\theta_k \sim p$, and construct dataset

$$\mathcal{D}_\phi = \left\{ \left( (s_1^m, a_{1:H}^m), \sum_{h \in [H]} \gamma^{h-1} \phi(s_h^m, a_h^m; \theta_k) \right) \right\}_{m \in [M], k \in [K]}.$$

4: Fit random Q-basis functions $\psi(\cdot, \cdot, \nu_k) : \mathcal{S} \times \mathcal{A}^H \to \mathbb{R}$ for $k \in [K]$ by minimizing the loss over the dataset $\mathcal{D}_\phi$,

$$\left\{ \nu_k^* \right\}_{k \in [K]} \in \operatorname*{argmin}_{\{\nu_k\}_{k \in [K]}} \quad \frac{1}{M} \sum_{m \in [M], k \in [K]} \left( \psi(s_1^m, a_{1:H}^m; \nu_k) - \sum_{h \in [H]} \gamma^{h-1} \phi(s_h^m, a_h^m; \theta_k) \right)^2.$$

5: **Online Planning Phase:**

6: Fit the testing task's reward function $r(\cdot, \cdot)$ with linear regression on random features:

$$w^* \in \operatorname*{argmin}_{w} \quad \frac{1}{MH} \sum_{h \in [H], m \in [M]} \left( r(s_h^m, a_h^m) - \sum_{k \in [K]} w_k \phi(s_h^m, a_h^m; \theta_k) \right)^2 + \lambda \|w\|_2^2$$

where $r(s_h^m, a_h^m) \sim R_{s_h^m, a_h^m}$.

7: Sample $s_1 \sim \mu_0$

8: **for** time index $t = 1, \cdots$ **do**

9:     Randomly generate $N$ sequences of actions $\{a_1^n, \cdots, a_H^n\}_{n \in [N]}$

10:     Find the best sequence such that

$$n_t^* \in \operatorname*{argmax}_{n \in [N]} \quad \sum_{k \in [K]} w_k^* \psi(s_t, a_{t:t+H-1}^n; \nu_k^*).$$

    Execute $a_t^{n_t^*}$ from the sequence $n_t^*$, observe the new state $s_{t+1} \sim \mathcal{T}(s_t, a_t^{n_t^*})$

11: **end for**

---

## B  ADDITIONAL EXPERIMENTS AND SETUP DETAILS

In this section, we provide more details of the experiments, including more detailed setup and supplementary results.

### B.1  DESCRIPTION OF ENVIRONMENTS

We describe the details of all used environments such as observation space, action space, reward, offline / online objectives, and dataset collection.

**Meta-World**  All our meta-world [52] domains share the standard meta-world observation which includes gripper location, and object locations of all possible objects involved in the Metaworld benchmark. Although the observation space has 39 dimensions, each domain only uses one or two objects so only 7 dimensions are changing any each domain we chose. For pixel observation variants of each domain, we concatenate two $84 \times 84 \times 3$ RGB images from two views, with a resulting

observation dimension of $42336$. Each domain has a 4 dimensional action space, corresponding to the delta movement of end-effector in the workspace along with the delta movement of the gripper finger. Metaworld provides a set of $50$ goal configurations for each domain. We collect offline dataset following the procedure described in Sec. B.2. The online objective is chosen to be a novel configuration that isn't any of the $50$ offline goal configurations. To create the privileged dataset, we choose $25$ of the goal configurations as offline objectives. These chosen configurations are the furthest $25$ goals from the online objective in the Euclidean distance. We evenly annotate the offline dataset with rewards for each of these goals to form a privileged dataset such that the online objective is out of the distribution of the offline objectives. For different configurations of the same domain, since object locations are in observation and the goal configuration isn't in it, the dynamics is the same. We now describe the objectives of all used meta-world domains, including those used in the appendix.

1. **Reach across Wall** The objective is to reach a target across a wall. The location of the target is randomized for each goal configuration.

2. **Open Door** The objective is to open the door to a certain angle by putting the end-effector between the door handle and the door surface. For each configuration, the location of the door is randomized.

3. **Turn on Faucet** The objective is to turn on a faucet by rotating it along an axis. For each goal configuration, the location of the faucet is randomized.

4. **Press Button** The objective is to press a button horizontally. The location of the button is randomized for each goal configuration.

**Hopper** Hopper is an environment with a higher dimensional observation of 11 dimensions and an action space dimension of 3. Hopper is a locomotion environment that requires long-horizon reasoning since a wrong action will make it fall down and touch the ground only after some steps. The objective of the original Hopper enviroment in OpenAI gym is to train it to run forward. To analyze the performance of CQL and SF on Hopper, we modify the objective to a goal-conditioned variant such that the agent is trained to follow a certain velocity specified by its goal. Similar modifications are common in meta reinforcement learning such as in [55]. We sample a set of $50$ goal velocities between $0$ and $1.0$ as offline objectives to collect data in the same way as we did in Metaworld environment. For online transfer, we choose four domains with four different online objectives. All the variant domains of Hopper share the same dynamics.

1. **Hopper Stand** The objective is to stand still and upright. This online objective is on the boundary of the offline objectives' range.

2. **Hopper Sprint** The objective is to run forward at a speed of $1.5$, which is out of the distribution of offline objectives ranging from $0$ to $1.0$. The reward function remains the same as that for offline objectives. Only the goal changes.

3. **Hopper Backward** The objective is to run backward at a target speed of $0.5$, which is in the opposite direction of the offline objective. Falling down to the ground is penalized.

4. **Hopper Jump** The objective is to jump to a height of $1.5$, without moving forward or backward. Such height is typically not achievable when running forward, so this objective is drastically different from offline objectives.

**D'Claw** D'Claw environment is a dexterous manipulation environment with 24 dimensions of observation space and 9 dimensions of action space. The 9 dimensions correspond to the 9 joints of a robot hand with 3 fingers and 3 joints on each finger. The objective of the environment is to control the hand to rotate a rotating tripod to a certain angle. Angular distance to the target angle is used as the offline objective. To increase the degree of freedom and make the environment more challenging, we allow the tripod to not only rotate but also translate freely on the 2d plane. The initial rotation and position of thetripod are randomized upon every episode. We collect the offline dataset in the same way as in meta-world, training on $50$ offline objectives and using $\epsilon$-greedy to collect rollouts. At test time we choose a new offline objective angle and annotate the rewards of the privileged dataset in the same way we did for goal conditioned Metaworld environments.

**Analytical 2D Point**   Point is a 2D point goal-reaching environment with linear dynamics. The reward is defined as the distance to goal minus an action penalty. The offline objectives are negative distances to the randomly selected goals on the 2d plane. The online objectives are novel goals on the plane. Since we are not evaluating CQL and SF on this environment, we don't generate the privileged dataset for 2D point.

**Point Perturbed**   Point Perturbed shares the same linear dynamics as Point, but features unsafe regions with negative rewards or local maxima with small positive rewards. These perturbations represent out-of-distribution test objectives that cannot be well approximated by a low-dimensional feature vector. Note that the added perturbations make point perturbed no longer an instance of analytical 2D point environment with a different goal. Instead, it features online objectives that are completely in a different class from 2D point.

### B.2   DESCRIPTION OF ALGORITHM TRAINING DETAILS

For each domain, we first train 50 policies with SAC [7]. Each policy is trained towards some offline objective of the domain described in Sec. B.1 for 50000 steps. We then use an $\epsilon$-greedy version of each trained policy to collect 32000 data points for each domain per offline objective. We choose $\epsilon = 0.5$. Such a procedure ensures the dataset has reasonable coverage of the entire state-action space. We note training these policies are fully optional, since `RaMP` only trajectories without rewards. Datasets collected via intrinsic rewards like curiosity would totally suffice. We choose the random feature dimension to be 2048. Each dimension in the random feature is extracted by feeding state-action tuple to a randomly initialized MLP with 2 hidden layers of size of 32. There are therefore 2048 independent, small random MLPs to extract $\phi$. All state-action tuples are projected to reward basis $\phi$ with it.

During offline training phase, we ensemble 8 instances of MLP with 2 hidden layers of size 4096 and train $\psi$ network following Sec. 3.1. We train $\psi$ network with a learning rate of $3 \times 10^{-4}$ on the offline dataset for 4 epochs, with a $\gamma$ decay of 0.99 and batch size 128. We choose the horizon $H$ to be 10 for meta-world and D'Claw environments and 32 for Hopper environments. During online adaptation phase, we first do random exploration for 2457 steps to collect enough data points for linear regression. When doing linear regression, we always concatenate a bias dimension to $\psi$. For each MPC rollout, we randomly sample 1024 action sequences. We penalize the predicted reward with 0.16 of the variance of predictions from all 8 ensembles. Since online least square makes recomputing $\omega$ regression fast, we perform update of weight vector every single step after initial random exploration is finished.

### B.3   ADDITIONAL RESULTS ON HOPPER STAND

Due to page limit, we omitted the plot for Hopper Stand in Fig.3. Here we provide additional results of `RaMP` and baselines for it in Fig. 4. The result is consistent with our analysis in the main paper. `RaMP` outperforms the baselines just in other Hopper variants. One major difference here is that CQL is performing well for HopperStand. This is likely because the online objective of Hopper Stand is at the boundary of offline objectives as described in Sec. B.1. Given that offline objectives are running at target velocities, the CQL likely learns to not fall down even if the online objective is out of distribution. By not falling down alone, CQL is capable of maintaining a good reward as seen in this case.

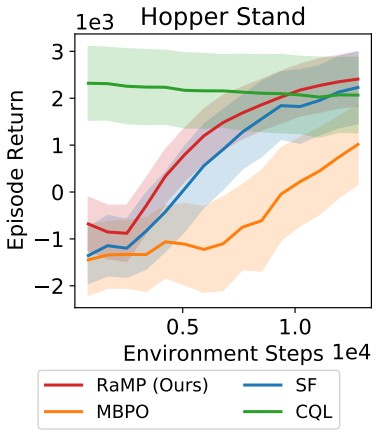

Figure 4: Results on Hopper Stand

### B.4   SCALING
TO HIGH-DIMENSIONAL PIXEL OBSERVATION

In Sec.4.3, we evaluate `RaMP`'s ability to scale to environments with high dimensional observations. In this section, we go a step further by significantly increasing the dimension of the observation space to 42336 as described in Sec. B.1. We use a CNN encoder following the architecture in [56] followed by 2 layer MLP as the random feature network.

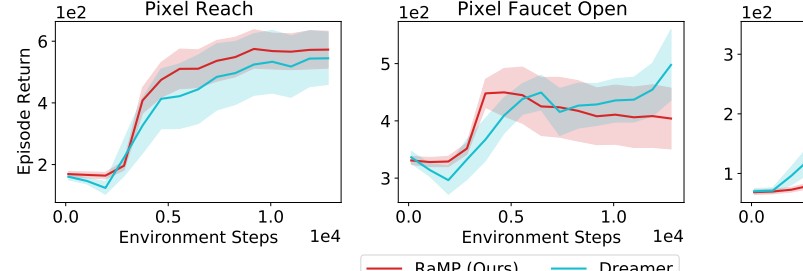

Figure 5: Results on Metaworld with high-dimensional pixel observation. `RaMP` achieves comparable performance to Dreamer on Pixel Reach and Pixel Faucet Open but struggles to perform well on Pixel Door Open. The performance of `RaMP` on Pixel Faucet Open and Pixel Reach is very similar to its performance in state observation environments.

Table 3: Return as a function of random feature dimension. Low dimensional random features are unable to approximate the true reward with linear regression, leading to degraded convergence performance.

|      | Point | Point Perturbed | Metaworld Reach |
|------|-------|-----------------|-----------------|
| 128  | $30.46 \pm 1.62$ | $29.61 \pm 2.55$ | $765.38 \pm 129.98$ |
| 256  | $31.53 \pm 1.45$ | $31.09 \pm 0.04$ | $806.12 \pm 104.77$ |
| 512  | $33.02 \pm 1.10$ | $\mathbf{33.13 \pm 2.09}$ | $824.56 \pm 121.52$ |
| 1024 | $33.36 \pm 1.10$ | $32.02 \pm 2.38$ | $\mathbf{855.14 \pm 85.23}$ |
| 2048 | $\mathbf{34.01 \pm 1.04}$ | $31.83 \pm 1.31$ | $830.84 \pm 142.02$ |
| 4096 | $33.31 \pm 1.15$ | $32.05 \pm 1.11$ | $802.45 \pm 94.47$ |

Both CNN and MLP layers are randomly initialized. Action is projected and concatenated to the first layer of MLP so the random feature would still be conditioned on both action and observation. We compare our method against the Dreamer [19], the state of art model-based learning method optimized for pixel observations. Similar to MBPO, we pre-train dreamer's dynamics branch with offline data before the online phase. As shown in Fig. 5, our method is able to achieve a similar level of performance as Dreamer in two meta-world environments, Pixel Reach and Pixel Faucet Open. `RaMP` does not see a significant return drop from the variant of the environment with state observation. Given the efficacy of Dreamer, the result still shows `RaMP`'s performance can scale up to pixel observation. However, Dreamer is able to outperform `RaMP` significantly in an environment like Pixel Door Open. This is likely because random features capture the change in input space rather than reward space. Pixel observations can give random features a lot of noise while important semantic features may not correspond to a big change in pixel space. We note that instead of using random convolution layers, we can use pre-trained encoders to achieve better semantic feature extraction and significantly improve the quality of random features. This is beyond the scope of our work and we leave this for future works.

### B.5 ADDITIONAL ABLATIONS

Our method builds on the assumption that a linear combination of high dimensional random features can approximate arbitrary reward functions. In the following ablation experiments, we validate this assumption both quantitatively and qualitatively.

**Effect of random feature dimension** We evaluate our method on Point, Point Perturbed, and Metaworld Reach using $\{128, 256, 512, 1024, 2048, 4096\}$ random features. Results are summarized in Table 3. We find that performance degrades with smaller random feature dimensions because the features are unable to linearly approximate the true reward. On the other hand, Fig. 6 shows that high dimensional random features experience slower adaptation.

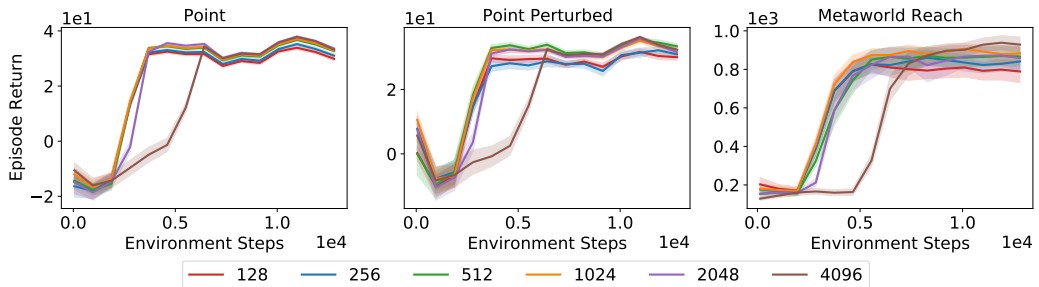

Figure 6: Learning curves for different random feature dimensions. Low-dimensional random features suffer from poor convergence performance, whereas high-dimensional random features experience slow adaptation.

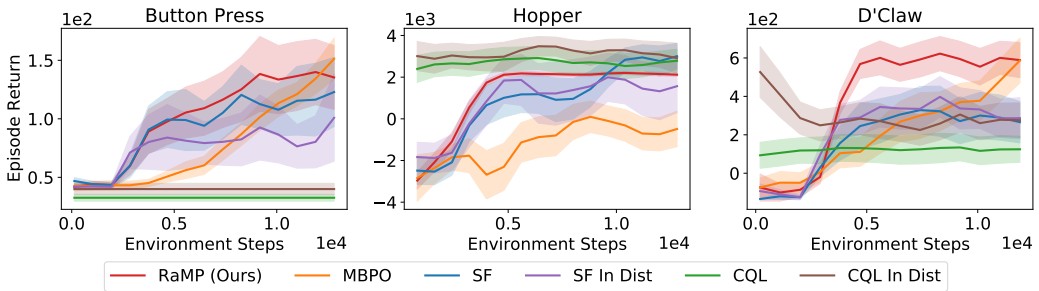

Figure 7: In-distribution results for Successor Features and CQL results. Note that `RaMP` and MBPO are unaffected since they do not depend on the distribution of the offline objectives.

**Scaling with state dimension** In Table 4 we evaluate our method on point reaching environments with 2, 3, and 4 state dimensions. All three environments feature distance-to-goal minus action penalty as the reward. We compute the return error stemming from linear regression as well as the Q error stemming from both linear regression and function approximation. We see an increase in both return error and Q error with higher state dimensions, but the overall approximation errors remain reasonably low.

**Nonlinear approximation capability** In Fig. 8, we visualize the truncated Q value obtained from a linear combination of the random cumulants and compare it to the ground truth Q value approximated by Monte-Carlo sampling. We perform the comparison with Point and Point Perturbed environments. Our method provides an accurate estimate of the Q value even in the face of out-of-distribution and highly nonlinear rewards.

Finally, we compare the performance of `RaMP` to baselines on in-distribution online objectives. Our method is designed to make no assumptions about test objectives during policy transfer. As shown in Fig. 3, `RaMP` outperforms CQL and SF when the online objectives are out of distribution. A natural question to ask is how things will change when the setting satisfies the assumptions of offline-online objective correlation. For example, in a 2D reaching environment, the training dataset may be annotated with either rewards corresponding to only goals on the right half or goals covering the entire plane.

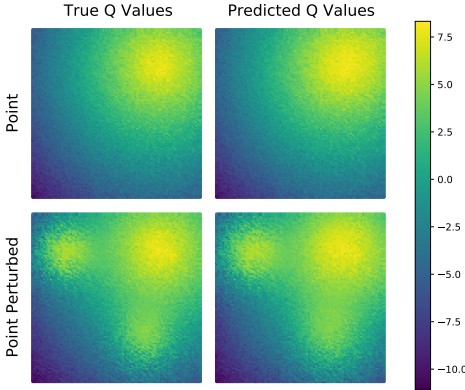

Figure 8: Visualization of true Q value and approximated Q value. Our method is able to approximate the Q value in the face of out-of-distribution and highly nonlinear rewards.

Table 4: Approximation error with different state dimensions. As state dimension increases, approximation error increases but remains in a reasonable range.

|  | Point 2D | Point 3D | Point 4D |
|---|---|---|---|
| Return Error | $0.42 \pm 0.20$ | $1.20 \pm 0.61$ | $1.32 \pm 0.70$ |
| Q Error | $0.32 \pm 0.05$ | $0.51 \pm 0.10$ | $0.61 \pm 0.12$ |

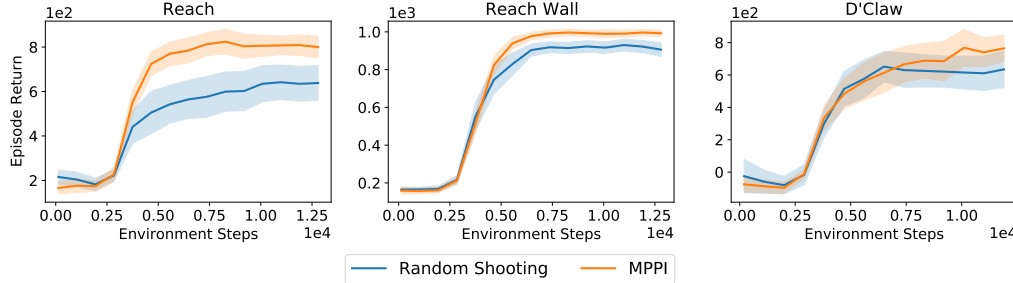

Figure 9: MPPI results on MetaWorld and D'Claw. MPPI improves the performance of our method across all four tasks, showing that our method can benefit from powerful planners.

When the online objective is to reach a goal on the left half of the plane, it will be out of distribution for the first case while being in distribution for the second. When we curate the labeling process of the privileged dataset to satisfy the in-distribution assumption, CQL and SF receive a significant performance boost. As shown in Fig. 7, the performance of our method and MBPO are unaffected as neither algorithm depends on offline objectives. CQL, on the other hand, matches the performance of our method under this new setting. This serves as a foil to the generalization of our method to out-of-distribution online objectives.

### B.6 MPPI EXPERIMENTS

We provide additional results on model-predictive control via model-predictive path integral (MPPI) [54] in Fig. 9. MPPI is an sampling-based trajectory optimization method which maintains a distribution of action sequence initialized as isotropic standard Gaussian. In each iteration, MPPI draws $n$ action sequences from the current distribution and computes their values, which we do using the learned Q-basis networks and online regression weights. MPPI then updates the distribution using the weighted mean and standard deviation of the sampled trajectories, where the weights are computed as the softmax of the values multiplied by a temperature parameter $\gamma$. As shown in Fig. 9, MPPI improves the performance of our method across two Metaworld environments and the D'Claw environment, thus indicating that our method can benefit from powerful planning algorithms. In these experiments, we perform 10 optimization steps and sample $n = 1000$ trajectories in each step. We use $\gamma = 10$ for Metaworld and $\gamma = 50$ for D'claw.

### B.7 FINETUNING EXPERIMENTS

While we freeze the Q-basis networks at test time in our main experiments to demonstrate transfer behavior, we can in fact finetune the Q-basis networks to continuously improve our estimate of the Q-value. After performing reward regression for a number of steps, we can finetune the Q-basis networks on online trajectories by fitting the predicted Q-values to the Monte-Carlo Q values and allowing the gradients to flow through the regression weights. We conduct finetuning experiments on two Metaworld environments and the D'Claw enironments. As shown in Fig. 10, our method sees a noticeable performance increase with finetuning starting at 6400 steps.

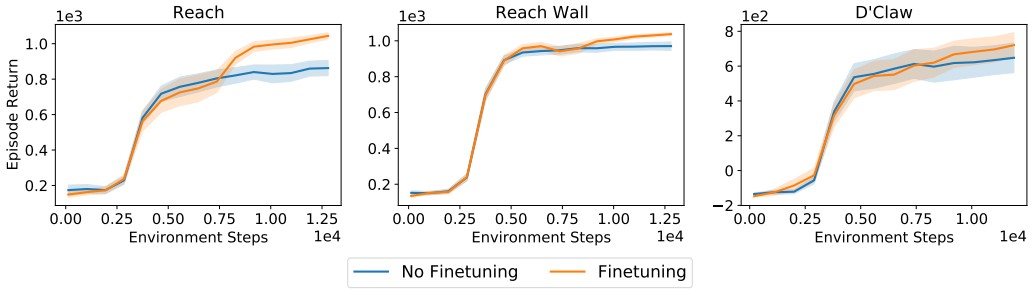

Figure 10: Finetuning results on MetaWorld and D'Claw. Our method is able to continuously improve by finetuning the Q-basis networks during online training.

## C    DETAILED THEORETICAL RESULTS

In this section, we provide the formal statement of the theoretical insights given in §3.3, and corresponding proofs.

### C.1    FORMAL STATEMENT

**Theorem C.1.** Suppose the offline data in $\mathcal{D}$ are generated from some distribution $\rho \in \Delta(\mathcal{S} \times \mathcal{A})$, i.e., $(s_h^m, a_h^m) \sim \rho(\cdot, \cdot)$ and $s_{h+1}^m \sim \mathcal{T}$ for all $(m, h) \in [M] \times [H]$, and $\inf_{(s,a)\in\mathcal{S}\times\mathcal{A}} \rho(s,a) = \underline{\rho} > 0$.

Suppose $\theta_k \sim p(\cdot)$ for all $k \in [K]$, and $\sup_{(s,a,\theta)\in\mathcal{S}\times\mathcal{A}\times\mathbb{R}^d} |\phi(s,a;\theta)| \leq \kappa$ for some $\kappa > 0$ and $\phi(\cdot,\cdot;\theta)$ is continuous. For some large enough $n := MH$, letting $\lambda = n^{-1/2}$, we have that if $K = \Omega(\sqrt{n}\log(\kappa^2\sqrt{n}/\delta))$, then with probability at least $1 - \delta$, for any given reward function $R$

$$\|\widehat{Q}_\pi(w^*) - Q_\pi\|_\infty \leq \frac{1}{1-\gamma}\sqrt{\frac{1}{\underline{\rho}}\left[\inf_{f\in\mathcal{H}}\underbrace{\sum_{(s,a)\in\mathcal{S}\times\mathcal{A}}\int\left(r - f(s,a)\right)^2 dR_{s,a}(r)\rho(s,a)}_{\mathcal{E}(f)} + \mathcal{O}\left(\frac{\log(1/\delta)}{\sqrt{n}}\right)\right]}$$

for any policy $\pi$, where $\mathcal{H} := \{f = \int \phi(\cdot,\cdot;\theta)w(\theta)dp(\theta) \mid \int |w(\theta)|^2 dp(\theta) < \infty\}$, $w^*$ is the solution to (3.2), and

$$\widehat{Q}_\pi(s, a; w^*) := \mathbb{E}\left[\sum_{h=1}^{\infty} \gamma^{h-1} \sum_{k\in[K]} w_k^* \phi(s_h, a_h; \theta_k) \,\Big|\, s_1 = s, a_1 = a\right]. \tag{C.1}$$

The proof of Theorem C.1 can be found in §C.2. It shows that with large enough amount of random features, the $Q$-function of any reward function $R$, under any policy $\pi$, can be approximated accurately up to some inherent error related to the richness of the function class that the features can represent. Note that we here only state the results under some mild and basic assumptions from the random features literature, and are by no means tight. They can be improved in various ways, for example, if the sampling distribution of $\theta$, $p$, can be data-dependent, and some stronger assumptions on the data and kernel function classes [47, 50].

**Corollary C.2.** Suppose the assumptions in Theorem C.1 hold, and additionally the kernel induced function space $\mathcal{H}$ is rich enough such that $\inf_{f\in\mathcal{H}} \mathcal{E}(f) = 0$. Then, with horizon length $H = \widetilde{\Theta}(\frac{\log(R_{\max}/\epsilon)}{1-\gamma})$ and $M = \widetilde{\Theta}(\frac{1}{(1-\gamma)^3\epsilon^4})$ episodes in dataset $\mathcal{D}$, and with $K = \widetilde{\Omega}((1-\gamma)^{-2}\epsilon^{-2})$ random features, we have $\|\widehat{Q}_\pi^H(w^*) - Q_\pi\|_\infty \leq \mathcal{O}(\epsilon)$ for any $\pi$, where for each $(s,a)$, $\widehat{Q}_\pi^H(s,a;w^*)$ is a $H$-horizon truncation of (C.1), which can be estimated from the offline dataset $\mathcal{D}$.

The proof of Corollary C.2 can be found in §C.2, which specifies the number of random features, the horizon length per episode, and the number of episodes, in order to approximate $Q_\pi$ accurately

by using the data in $\mathcal{D}$. Note that the number of random features is not excessive and is polynomial in problem parameters. Combining Theorem C.1 and Corollary C.2 leads to the informal statement in Theorem 3.1.

Next, we justify the use of open-loop $Q$-functions in planning in a deterministic transition environment, which contains all the environments our empirical results we have evaluated. Recall that for any given reward $R$, let $Q_\pi$ denote the $Q$-function under policy $\pi$. Note that with a slight abuse of notation, $Q_\pi$ can also be the multi-step $Q$-function (see definition in §2), and the meaning should be clear from the input, i.e., whether it is $Q_\pi(s, a)$ or $Q_\pi(s, a_1, \cdots, a_H)$.

**Theorem C.3.** Let $\Pi$ be some subclass of Markov stationary policies, i.e., for any $\pi \in \Pi$, $\pi : \mathcal{S} \to \Delta(\mathcal{A})$. Suppose the transition dynamics $\mathcal{T}$ is deterministic. For any given reward $R$, denote the $H$-step policy obtained from $H$-step open-loop policy improvement over $\Pi$ as $\pi'_H : \mathcal{S} \to \mathcal{A}^H$, defined as

$$\pi'_H(s) \in \underset{(a_1, \cdots, a_H) \in \mathcal{A}^H}{\operatorname{argmax}} \max_{\pi \in \Pi} Q_\pi(s, a_1, \cdots, a_H),$$

for all $s \in \mathcal{S}$. Finally, define the value-function under $\pi'_H$ as $V_{\pi'_H}(s) := Q_{\pi'_H}(s, \pi'_H(s))$, where $Q_{\pi'_H}(s, a_{1:H})$ is the fixed-point of the Bellman operator $\mathcal{T}_{H,\pi'_H}$ defined in (C.8). Then, we have that for all $s \in \mathcal{S}$

$$V_{\pi'_H}(s) \geq \max_{a_{1:H}} \max_{\pi \in \Pi} Q_\pi(s, a_1, \cdots, a_H) \geq \max_a \max_{\pi \in \Pi} Q_\pi(s, a).$$

## C.2 DEFERRED PROOFS

### C.2.1 PROOF OF THEOREM C.1

The proof relies on the result of generalization guarantees of learning from random features, with squared loss. Note that one cannot use the results in [46], which dealt with Lipschitz loss function of the form $c(y', y) = c(y'y)$. This does not include the squared loss we used in our experiments. Instead, we resort to the results in [47], which also yield a better statistical rate. For the sake of completeness, we re-state the abridged version of one key result therein, Theorem 1 in [47], as follows.

**Lemma C.4.** Suppose that $K$ is a kernel with an integral representation $K(x, x') = \int_\Omega \psi(x, w)\psi(x', w)dp(w)$, where $(\Omega, p)$ is a probability space and $\psi : X \times \Omega \to \mathbb{R}$, where $X$ is a separable space. Suppose $\psi$ is continuous and $|\psi(x, w)| \leq \kappa$ with $\kappa \in [1, +\infty)$ almost surely, and $|y| \leq b$ almost surely. Define the expected risk:

$$\mathcal{E}(f) := \int (f(x) - y)^2 d\rho(x, y),$$

where $\rho$ is the distribution where the data samples $(x_i, y_i)_{i=1}^n$. Define the solution to kernel ridge regression with $M$ random features as

$$\widehat{f}_{\lambda,M}(x) = \phi_M(x)^\top \widehat{w}_{\lambda,M}, \quad \text{with} \quad \widehat{w}_{\lambda,M} := (\widehat{S}_M^\top \widehat{S}_M + \lambda \boldsymbol{I})^{-1} \widehat{S}_M^\top \widehat{y}, \tag{C.2}$$

where $\phi_M(x) := \big(\psi(x, w_1), \psi(x, w_2), \cdots, \psi(x, w_M)\big)/\sqrt{M}$, $w_i$ are drawn i.i.d. from $p(\cdot)$, $\widehat{y} := (y_1, \cdots, y_n)/n^{1/2}$, $\widehat{S}_M^\top := \big(\phi_M(x_1), \cdots, \phi_M(x_n)\big)/n^{1/2}$. Then, suppose $n \geq n_0$, $\lambda = 1/n^{1/2}$, and the number of features $M \geq c_0\sqrt{n} \log(108\kappa^2\sqrt{n}/\delta)$, we have that with probability at least $1 - \delta$,

$$\mathcal{E}(\widehat{f}_{\lambda,M}) - \min_{f \in \mathcal{H}} \mathcal{E}(f) \leq \frac{c_1 \log^2(18/\delta)}{\sqrt{n}},$$

where $n_0, c_0, c_1$ are absolute constants, $\mathcal{H}$ is the reproducing kernel Hilbert space corresponding to the kernel $K$.

We then apply Lemma C.4, with $(x, y)$ in the lemma being replaced by $\big((s, a), r(s, a)\big)$, $\rho(x, y), p, x, w, M, \lambda$ in the lemma being replaced by $\rho(s, a) \cdot R_{s,a}, p, (s, a), \theta, K, \lambda$ in our case.

Note that Lemma C.4 requires the space $X$ to be separable, and our finite space $\mathcal{S} \times \mathcal{A}$ satisfies; it requires $|y|$ bounded, and our reward is absolutely bounded by $R_{\max}$, and thus also satisfies. We hence obtain that with probability at least $1 - \delta$, if the number of random features $K \geq \Omega(\sqrt{n}\log(\sqrt{n}))$, with $n := HM$, then

$$\mathbb{E}_{(s,a)\sim\rho(\cdot,\cdot),r\sim R_{s,a}(\cdot)}\left(r - \sum_{k\in[K]} w_k^*\phi(s,a;\theta_k)\right)^2 \leq \inf_{f\in\mathcal{H}} \mathcal{E}(f) + \mathcal{O}\left(\frac{\log(1/\delta)}{\sqrt{n}}\right) \qquad \text{(C.3)}$$

where we note that $w^* = (w_1^*, \cdots, w_K^*)$ is the solution to (3.2), and the $\mathcal{E}(f)$ here is defined in Theorem C.1.

For any policy $\pi$ for the MDP, let $Q_\pi$ denote the $Q$-function under policy $\pi$ and the actual reward function distribution $R$, and $\widehat{Q}_\pi(w^*)$ denote the $Q$-function under the estimated reward using random features:

$$\widehat{Q}_\pi(s,a;w^*) := \mathbb{E}\left[\sum_{h=1}^\infty \gamma^{h-1}\widehat{r}(s_h,a_h;w^*) \,\Big|\, s_1 = s, a_1 = a\right],$$

$$\text{where} \quad \widehat{r}(s,a;w^*) := \sum_{k\in[K]} w_k^*\phi(s,a;\theta_k). \qquad \text{(C.4)}$$

By Bellman equation, we have that for each $(s,a)$

$$\left|Q_\pi(s,a) - \widehat{Q}_\pi(s,a;w^*)\right| = \left|\int r\,dR_{s,a}(r) + \gamma\sum_{s',a'} Q_\pi(s,a)\mathcal{T}(s'\,|\,s,a)\pi(a'\,|\,s')\right.$$

$$\left. - \widehat{r}(s,a;w^*) - \gamma\sum_{s',a'} \widehat{Q}_\pi(s,a;w^*)\mathcal{T}(s'\,|\,s,a)\pi(a'\,|\,s')\right|$$

$$\leq \left|\int r\,dR_{s,a}(r) - \widehat{r}(s,a;w^*)\right| + \gamma\cdot\left\|Q_\pi - \widehat{Q}_\pi(w^*)\right\|_\infty.$$

Taking sup over $s, a$ and organizing the terms, we have

$$\left\|Q_\pi - \widehat{Q}_\pi(w^*)\right\|_\infty \leq \frac{1}{1-\gamma}\cdot\sup_{s,a}\left|\int r\,dR_{s,a}(r) - \widehat{r}(s,a;w^*)\right|$$

$$\leq \frac{1}{1-\gamma}\cdot\sqrt{\sum_{s,a}\left(\int r\,dR_{s,a}(r) - \widehat{r}(s,a;w^*)\right)^2} \qquad \text{(C.5)}$$

$$\leq \frac{1}{1-\gamma}\cdot\sqrt{\frac{1}{\underline{\rho}}\sum_{s,a}\left(\int r\,dR_{s,a}(r) - \widehat{r}(s,a;w^*)\right)^2\rho(s,a)} \qquad \text{(C.6)}$$

$$= \frac{1}{(1-\gamma)\sqrt{\underline{\rho}}}\cdot\sqrt{\mathbb{E}_{(s,a)\sim\rho(s,a)}\left(\int r\,dR_{s,a}(r) - \widehat{r}(s,a;w^*)\right)^2} \qquad \text{(C.7)}$$

where (C.5) uses that $\|\cdot\|_\infty \leq \|\cdot\|_2$ for finite-dimensional vectors, (C.6) uses the definition of $\underline{\rho}$. Further, by Jensen's inequality, for each $(s,a)$

$$\left(\int r\,dR_{s,a}(r) - \widehat{r}(s,a;w^*)\right)^2 = \left(\mathbb{E}_{r\sim R_{s,a}(\cdot)}\left[r - \widehat{r}(s,a;w^*)\right]\right)^2 \leq \mathbb{E}_{r\sim R_{s,a}(\cdot)}\left[r - \widehat{r}(s,a;w^*)\right]^2,$$

which, combined with (C.7) and (C.3), gives that

$$\left\|Q_\pi - \widehat{Q}_\pi(w^*)\right\|_\infty \leq \frac{1}{(1-\gamma)\sqrt{\underline{\rho}}}\cdot\sqrt{\mathbb{E}_{(s,a)\sim\rho(s,a),r\sim R_{s,a}(\cdot)}\left(\int r\,dR_{s,a}(r) - \widehat{r}(s,a;w^*)\right)^2}$$

$$\leq \frac{1}{(1-\gamma)\sqrt{\underline{\rho}}}\cdot\sqrt{\inf_{f\in\mathcal{H}} \mathcal{E}(f) + \mathcal{O}\left(\frac{\log(1/\delta)}{\sqrt{n}}\right)},$$

which completes the proof. $\qquad\square$

### C.2.2 PROOF OF COROLLARY C.2

First, note that with $H = \Theta(\frac{\log(R_{\max}/(\epsilon(1-\gamma)))}{1-\gamma})$ ensures that $\|\widehat{Q}_\pi^H(w^*) - \widehat{Q}_\pi(w^*)\|_\infty \leq \mathcal{O}(\epsilon)$, which can be obtained by the boundedness of $r(s,a)$ by $R_{\max}$, and the fact that

$$\gamma^H \frac{R_{\max}}{1-\gamma} = (1-(1-\gamma))^{\frac{1}{1-\gamma} \cdot H(1-\gamma)} \frac{R_{\max}}{1-\gamma} \leq \left(\frac{1}{e}\right)^{\log(R_{\max}/(\epsilon(1-\gamma)))} \frac{R_{\max}}{1-\gamma} = \epsilon.$$

Furthermore, since Theorem C.1 requires $n = HM = \widetilde{\Theta}(\frac{1}{(1-\gamma)^4 \epsilon^4})$, to make sure $\|\widehat{Q}_\pi(w^*) - Q_\pi\|_\infty \leq \epsilon$. Combining these facts yields the desired result. $\square$

### C.2.3 PROOF OF THEOREM 3.2

Define

$$Q_H^{\max}(s, a_1, \cdots, a_H) := \max_{\pi \in \Pi} Q_\pi(s, a_1, \cdots, a_H), \quad \text{and} \quad Q^{\max}(s,a) := \max_{\pi \in \Pi} Q_\pi(s,a).$$

We also define the Bellman operator under the open-loop policy $\pi_H'$ as follows: for any $Q \in \mathbb{R}^{|\mathcal{S}| \times |\mathcal{A}^H|}$,

$$\mathcal{T}_{H,\pi_H'}(Q)(s, a_1, \cdots, a_H) = \mathbb{E}\left[\sum_{h \in [H]} \gamma^{h-1} r(s_h, a_h) + \gamma^H Q(s_{H+1}, \pi_H'(s_{H+1})) \,\Big|\, s_1 = s, a_{1:H}\right].$$
(C.8)

Note that $\mathcal{T}_{H,\pi_H'}$ is a contracting operator, and we denote the fixed point of the operator as $Q_{H,\pi_H'} \in \mathbb{R}^{|\mathcal{S}| \times |\mathcal{A}^H|}$, which is the $Q$-value function under open-loop policy $\pi_H'$. By definition, we also know that the state-value function under $\pi_H'$, $V_{H,\pi_H'} = Q_{H,\pi_H'}(s, \pi_H'(s))$, i.e., by applying the open-loop policy $\pi_H'$ to the actions $a_{1:H}$ in $Q_{H,\pi_H'}(s, a_{1:H})$.

Note that

$$\mathcal{T}_{H,\pi_H'}(Q_H^{\max})(s, a_1, \cdots, a_H) = \mathbb{E}\left[\sum_{h \in [H]} \gamma^{h-1} r(s_h, a_h) + \gamma^H Q_H^{\max}(s_{H+1}, \pi_H'(s_{H+1})) \,\Big|\, s_1 = s, a_{1:H}\right]$$

$$= \mathbb{E}\left[\sum_{h \in [H]} \gamma^{h-1} r(s_h, a_h) + \gamma^H \max_{a_{H+1:2H}} Q_H^{\max}(s_{H+1}, a_{H+1:2H}) \,\Big|\, s_1 = s, a_{1:H}\right] \qquad \text{(C.9)}$$

$$\geq \mathbb{E}\left[\sum_{h \in [H]} \gamma^{h-1} r(s_h, a_h) + \gamma^H \max_{a_{H+1:2H}} Q_\pi(s_{H+1}, a_{H+1:2H}) \,\Big|\, s_1 = s, a_{1:H}\right] \qquad \text{(C.10)}$$

$$\geq \mathbb{E}\left[\sum_{h \in [H]} \gamma^{h-1} r(s_h, a_h) + \gamma^H Q_\pi(s_{H+1}, \pi(s_{H+1}) \cdots, \pi(s_{2H})) \,\Big|\, s_1 = s, a_{1:H}\right], \qquad \text{(C.11)}$$

$$= Q_\pi(s, a_1, \cdots, a_H), \qquad \text{(C.12)}$$

for any $\pi \in \Pi$, where (C.9) uses the definition of $\pi_H'$, (C.10) is due to the definition of $Q_H^{\max}$, and (C.11) is by the $\max_{a_{H+1:2H}}$, and (C.12) is by definition. Since (C.12) holds for any $\pi \in \Pi$, by the monotonicity of $\mathcal{T}_{H,\pi_H'}$, we have

$$Q_{H,\pi_H'}(s, a_{1:H}) = \lim_{k \to \infty} (\mathcal{T}_{H,\pi_H'})^k (Q_H^{\max})(s, a_{1:H}) \geq Q_H^{\max}(s, a_{1:H}) \geq \max_{\pi \in \Pi} Q_\pi(s, a_{1:H}).$$
(C.13)

Notice that for all $s$, by applying $\pi_H'(s)$ on both sides of (C.13),

$$V_{H,\pi_H'}(s) = Q_{H,\pi_H'}(s, \pi_H'(s)) \geq \max_{\pi \in \Pi} Q_\pi(s, \pi_H'(s)) = \max_{a_{1:H}} \max_{\pi \in \Pi} Q_\pi(s, a_{1:H}). \qquad \text{(C.14)}$$

Further, due to the multi-step maximization, we have

$$\max_{a_{1:H}} \max_{\pi \in \Pi} Q_\pi(s, a_1, \cdots, a_H) \geq \max_a \max_{\pi \in \Pi} Q_\pi(s, a),$$

which, combined with (C.14), completes the proof. $\square$

# D    RELATIONSHIP TO EXISTING WORK

We briefly connect our proposed algorithm to prior work.

**Successor features for transfer in RL:**    While successor features [40, 44] have shown the ability to transfer across problems in RL, the two key differences in our framework are (1) using random features rather than requiring learned or pre-provided features and (2) training open-loop Q functions rather than typical $Q_\pi$. These two changes allow transfer to happen across a broader class of reward functions and not simply be restricted to the policy cover experienced at training time.

**Model-based RL:**    Our work is connected to model based RL in that it disentangles dynamics and rewards, but is crucially different in that it doesn't model one-step evolution of state but rather long term accumulation of random features. This trades off compounding error for generalization error.

**Model-free RL:**    Our work is connected to methods for model-free RL in that it also models a set of Q-functions, but importantly this doesn't correspond to a particular reward, but rather to random features of state. By doing so, we are able to adapt to arbitrary rewards at test-time, rather than being tied to a particular reward function.

# E    INFINITE-HORIZON $Q$-FUNCTION VARIANT

While our setup in Section 2 uses a finite-horizon $Q_\pi^H$ to approximate $Q_\pi$, our method can also plan with an infinite-horizon $Q$ function during the online phase. In this section, we describe one compatible way to learn an infinite-horizon $Q$-function while still enjoying the benefits of RaMP in the case with *deterministic* transition dynamics. We also present empirical results and analysis of this variant.

## E.1    METHOD

We first notice that an infinite-horizon $Q$-function $Q_\pi$ can be decomposed into the discounted sum of an $H$-step reward and a discounted value function under policy $\pi$ evaluated at $s_{t+H}$:

$$Q_\pi(s', a') = \mathbb{E}_{\substack{a_t \sim \pi(\cdot \mid s_t) \\ s_{t+1} \sim \mathcal{T}(\cdot \mid s_t, a_t)}} \left[ \gamma^H V_\pi(s_{H+1}) + \sum_{t=1}^{H} \gamma^{t-1} r(s_t, a_t) \,\Big|\, s_1 = s', a_1 = a' \right]$$

$$\text{where } V_\pi(s') = \mathbb{E}_{a_t \sim \pi(\cdot \mid s_t)} \left[ \sum_{t=1}^{\infty} \gamma^{t-1} r(s_t, a_t) \,\Big|\, s_1 = s' \right].$$

Given a policy $\pi$, value function $V_\pi^\theta$ parameterized by $\theta$ can be learned via gradient descent and Monte-Carlo method:

$$\theta \leftarrow \theta - \alpha \nabla_\theta \, ||V_\pi^\theta(s_t) - (r_t + \gamma V_\pi^{\theta'}(s_{t+1}))||_2^2 \, , \text{ for sampled } (s_{t:t+1}, a_t, r_t) \sim \tau_\pi$$

where $\tau_\pi$ is trajectory rollouts collected with current policy $\pi$ and $\theta'$ is a target network that gets updated by $\theta$ with momentum.

Now consider our multi-step setup. Our multi-step $Q$-function can also be written as the sum of our $H$ step approximation and discounted value function at $s_{H+1}$:

$$Q_\pi(s, \widetilde{a}_{1:H}) = \mathbb{E}_{\substack{a_{H+t} \sim \pi(\cdot \mid s_{H+t}) \\ s_{t+1} \sim \mathcal{T}(\cdot \mid s_t, a_t)}} \left[ \sum_{t=1}^{\infty} \gamma^{t-1} r_t \,\Big|\, s_1 = s, a_1 = \widetilde{a}_1, \cdots, a_H = \widetilde{a}_H \right]$$

$$= Q_\pi^H(s, \widetilde{a}_{1:H}) + \gamma^H V_\pi(s_{H+1}),$$

where we note that in the last line, there is no expectation over $s_{H+1}$ since the transition dynamics is deterministic, and $s_{H+1}$ is deterministically decided by $(s_1, a_{1:H})$.

Vanilla RaMP enables efficient estimation of $Q_\pi$ with novel reward function at the cost of truncating the second term above with $Q_\pi \approx Q_\pi^H$. As we have shown in Section 4, planning with this finite-horizon $Q$-approximation would already lead to reasonable planning in most of the experiments.

We can go a step further and also estimate the second term so we can plan in an infinite-horizon setting. The main challenge is getting $V_\pi(s_{H+1})$ in our multi-step setup, as we don't explicitly predict $s_{H+1}$. This, however, can be addressed easily by reparameterizing $V_\pi(s_{H+1})$ on an action sequence that leads to $s_{H+1}$ just like what we did for $Q$. We thus define a multi-step value function

$$F_\pi(s, \widetilde{a}_{1:H}) = \mathbb{E}_{s_{H+1}}\Big[V_\pi(s_{H+1}) \,\big|\, s_1 = s, a_1 = \widetilde{a}_1, \cdots, a_H = \widetilde{a}_H\Big].$$

Then $Q_\pi(s, a_{1:H}) = Q_\pi^H(s, a_{1:H}) + \gamma^H \cdot F_\pi(s, a_{1:H})$. Under our deterministic transition dynamics, $s_{H+1}$ is fully determined by $(s_1, \widetilde{a}_{1:H})$, so we can remove the expectation in the equation. We then rewrite the training objective $V_\pi$ in terms of $F_\pi$ to learn $F_\pi$:

$$\theta \leftarrow \theta - \alpha \nabla_\theta \, \|V_\pi^\theta(s_{t+H}) - (r_{t+H} + \gamma V_\pi^{\theta'}(s_{t+H+1}))\|_2^2$$
$$\text{for sampled } (s_{t+H:t+H+1}, a_{t+H}, r_{t+H}) \sim \tau_\pi$$

becomes

$$\theta \leftarrow \theta - \alpha \nabla_\theta \, \|F_\pi^\theta(s_t, a_{t:t+H-1}) - (r_{t+H} + \gamma F_\pi^{\theta'}(s_{t+1}, a_{t+1:t+H}))\|_2^2$$
$$\text{for sampled } (s_{t:t+1}, a_{t:t+H}, r_{t+H}) \sim \tau_\pi$$

where we parameterize multi-step value function $F_\pi$ by $F_\pi^\theta$. So we can learn $F_\pi$ with Monte-Carlo sampling and gradient descent just like what we do to learn $V_\pi$ in the single-step case above. Combined with $Q_\pi^H$, we now have an estimation for the infinite-horizon $Q$-function $Q_\pi$ in a multi-step manner.

For planning, we do on policy learning by alternating between policy rollout and $Q_\pi$ learning. As a policy, MPC planner first uses the infinite-horizon $Q_\pi(s, a_{1:H}) = Q_\pi^H(s, a_{1:H}) + \gamma^H \cdot F_\pi(s, a_{1:H})$ to plan and collect rollouts. Then $F_\pi$ is trained with these on policy rollouts while $Q_\pi^H$ is also learned like vanilla RaMP via online least square. By incorporating this infinite-horizon variant, our MPC planner can now plan with an infinite horizon during the online phase.

### E.2 Experiment and Analysis

We implemented the infinite-horizon variant described above and carried out experiments to quantitatively evaluate its effect on performance in four environments with varying types of tasks. As shown in Figure 11, we found that the infinite-horizon variant out-performs vanilla RaMP with finite horizon $Q$-function in Hopper Jump and achieves comparable performance in D'Claw, both being the longer horizon environments among the four. However, the infinite-horizon variant actually performs worse in two Metaworld manipulation environments, ReachWall and FaucetOpen, likely because these tasks do not require infinite-horizon reasoning. In particular, the multi-step value function $V$ has to be learned from scratch from samples, which may even hurt the performance at the beginning of the online phase.

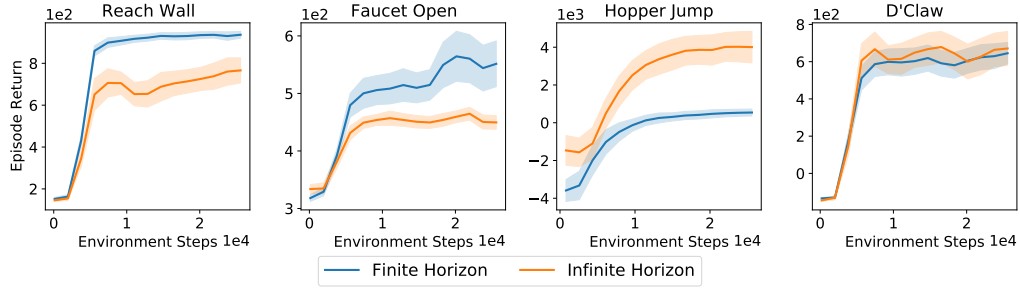

Figure 11: Results of RaMP variant with infinite horizon $Q$-function on four different environments. We notice this variant leads to better performance on longer-horizon tasks but doesn't increase the performance of metaworld tasks, likely because the benchmarked tasks don't require infinite-horizon reasoning. Since the $V$-function has to be learned from scratch, it could also hurt the performance at the beginning in our low data setting.

We also present results of this variant with infinite-horizon $Q$-function on all 8 environments compared with all baselines. We run all the environments for 25600 steps, at twice the amount of steps

we did in Section 4. We also changed the discount factor $\gamma$ to $0.9$ to emphasize the effectiveness of RaMP's quick adaptation to new reward using $Q_\pi^H$. As shown in Figure 12, RaMP has the ability to quickly adapt to a new reward function, when the amount of data is extremely low. As the algorithm sees more data, the advantage of RaMP is not salient as our infinite-horizon value function learning is using vanilla bootstrapping. However, since the infinite horizon-variant uses bootstrapping, the method can continuously improve like all RL methods as samples grow. This will be useful for longer horizon and harder tasks.

We note that although we used the simplest bootstrapping approach to learn multi-step $V$, we can use any other value estimation method to make it more efficient. The core of this variant is to parameterize the value function at the $H + 1$-step with the initial state and a sequence of actions.

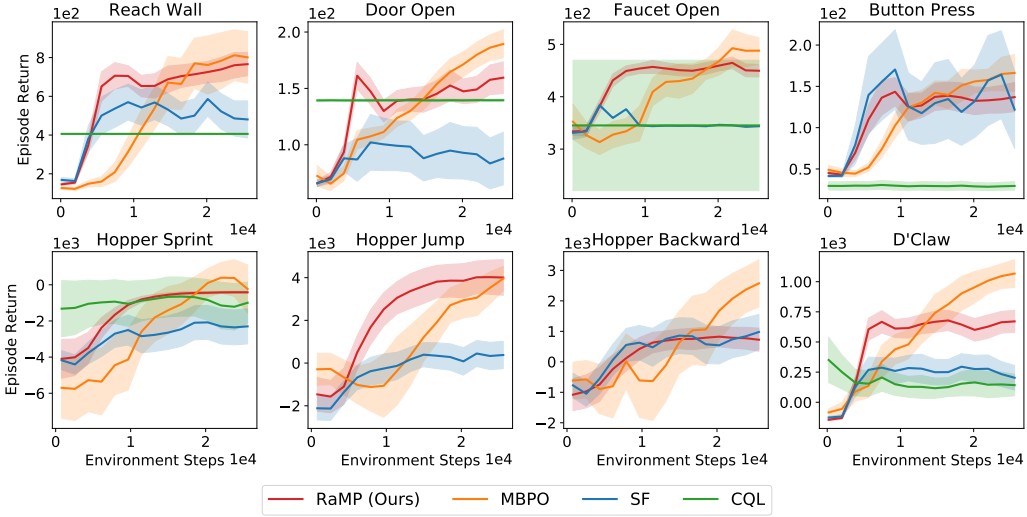

Figure 12: We benchmark RaMP with infinite-horizon $Q$-function on all environments for more steps. RaMP can quickly adapt to new test rewards in low-data regime and can continuously improve with the addition of bootstrapping. However, it will need to rely on more-sample efficient value function learning methods to be more sample efficient in the infinite-horizon setting. This shows RaMP's benefits largely lie in the low-data regime

## F    COMPLEXITY ANALYSIS

The space complexity of RaMP is primarily determined by the number of random features that are needed. As we describe in Corollary C.2, we require $K = \widetilde{\Omega}((1 - \gamma)^{-2}\epsilon^{-2})$ random features to achieve $\epsilon$-order error between the estimated and true $Q$-function. As the required approximation error decreases, the space needed to store all the features grows sublinearly. Note that this result is for *any* given reward function (including the target one) under *any* policy $\pi$, but not tied to a specific one (under the realizability assumption of the reward function).

On the other hand, the space complexity of the classical successor feature method is relatively *fixed* in the above sense: the dimension of the feature is fixed and does not change with the accuracy of approximating the optimal Q-function for the target task. However, the resulting guarantee is also more restricted: it is only for the optimal Q-function of the specific target reward, and also depends on the distance between the previously seen and the target reward functions. Hence, the two space complexities are not necessarily comparable.

