# OpenReview forum: "Model-free Reinforcement Learning that Transfers Using Random Reward Features"
_ICLR.cc/2023/Conference — Submitted to ICLR 2023_

### Official Review · Reviewer_R3oi · 2022-10-24

**Confidence:** 2
**Correctness:** 3
**Technical Novelty And Significance:** 3
**Empirical Novelty And Significance:** 3
**Recommendation:** 5

**Clarity, Quality, Novelty And Reproducibility:**

The paper is well-written. The method builds on other works but introduces novel elements and places in another context.


**Strength And Weaknesses:**

Strengths:
- well-motivated and important problem
- sound method

Weaknesses:
- scalability/applicability
- empirical part

**Summary Of The Paper:**

The paper proposes a method of transfer based on the assumption of common dynamics and variable reward signals. The method leverages artificial pseudo-rewards to span a space of probably $Q$-functions related to the down-stream tasks.


**Summary Of The Review:**

The problem studied in the paper is well-motivated and important. The authors develop a method, which takes advantage of the factorization of the dynamics and rewards. The premise is that the dynamics can be transferred and adaptation to a particular reward structure can be achieved relatively fast.

The authors propose an implicit method of encoding the model, via random cumulants in order to avoid the usual nuisance of predictive model-based approaches, like compounding errors. This, however, leads to the question, which is my main concern. The approach assumes training a multistep $Q$-function, $Q(a_1, \ldots, a_H)$ -- n the paper, $H\in \{10, 32\}$. I have doubts if this can be done efficiently for larger $H$, namely, when the long-term credit assignment is  needed. I suspect that either the method requires rather dense and informative rewards.

Questions:
 - Could the authors comment on implicit assumptions for the method to work (e.g. if I am right with the above concerns)? What kind of problems do have such properties?
 - Have the authors tries other optimization methods, e.g. cross-entropy method.

---

### Official Review · Reviewer_xLea · 2022-10-24

**Confidence:** 3
**Correctness:** 2
**Technical Novelty And Significance:** 3
**Empirical Novelty And Significance:** 3
**Recommendation:** 3

**Clarity, Quality, Novelty And Reproducibility:**

Clarity is relatively good, aside from a few important missing details.

I believe the presented theory contains errors which hamper the quality.

As far as I know, the approach is quite original and fairly interesting, though I am not that familiar with the relevant literature on transfer learning.

Reproducibility is not great given certain omitted details.

**Strength And Weaknesses:**

Strengths
=========
The method presented in this paper is quite simple and interesting as a reasonable baseline for the transfer RL setting. The idea of learning action values for a large number of random reward functions as a tractable route towards transfer to new environments is intriguing.

For the most part, the proposed method is coherent, reasonable and well-described.

The writing of the paper is generally good.


Weaknesses and Uncertainties
============================
I am not sure Theorem 3.1 is correct. In particular, there is no stated assumption on the distribution of the random features either in the main body or in the appendix. In the trivial case where all the random features are the same, clearly, the result would not hold as drawing multiple copies of the same feature would not help to represent the function any better. The proof in the appendix apparently uses Theorem 1 of Rudi and Rosaco (2017), however, that result is specifically for random features which approximate some Kernel, which I can't see is the case here, so it's not at all clear to me whether this result is relevant or applicable. Perhaps the authors or one of the other reviewers can clarify this as I am not personally very familiar with all the relevant work.

I also have doubts about Theorem 3.2, especially the part that states that the value of following an open-loop policy is greater than or equal to that of a one-step policy. In the case where $\Pi$ is the set of all possible policies, it contains the optimal policy, which cannot be improved upon. On the other hand, it is clear that with stochastic transitions there are cases where committing to an open-loop policy for more than one step is detrimental as it deprives you of the ability to choose actions in response to future random transitions for some horizon. The first inequality of the theorem appears to follow immediately from the definition of $V_{\pi'_H}$ and can be strengthened to equality unless I've misread something, so I also don't see how this part serves to justify the use of open-loop Q-functions in planning. Perhaps the result is meant to apply to deterministic transition dynamics only, but this is not stated anywhere and the background section makes it clear that stochastic transitions are considered.

It seems like all the displayed results are for a fairly low number of environment steps (around 10,000, see Figure 3), and it appears that in a few cases MBPO may surpass the performance of the proposed method given a little more time. While there is no inherent problem with arguing that the proposed method is useful in the low data regime, it would nonetheless be good to show how the different approaches compare given more updates. Furthermore, no details are given about how the hyperparameters of the approaches are selected which is crucially important in understanding how these results should be interpreted. Especially in the low data regime, things like the learning rate will have a large effect on the performance of different approaches after a fixed number of steps. In the same vein, I wonder if the performance of MBPO could be improved simply by adding more model rollouts.

If my understanding is correct, this approach selects action sequences which maximize reward only for the duration of the action sequence itself (fixed to 10 for metaworld and 32 for hopper). This is limiting, as one would have to explicitly predict the sum of reward conditioned on any possible action sequence for the entire horizon. Moreover, there seems to be no easy way to generalize the method beyond these circumstances, since any maximization in the random Q-functions would break the linearity which allows one linearly map rewards to associated action-sequence values, however, the paper does explain this point fairly clearly. For the same reason, any significant amount of stochasticity has the potential to break the method, which I didn't find was clearly explained in the paper, and in fact, the theory discussed above seems to incorrectly imply the opposite.

I'm a little confused about exactly how RaMP learns the reward function given a new environment. In appendix A it seems to be suggested that it queries the new reward function on the existing transition data from the offline dataset in order to gather data to train its reward prediction. If so, this is not really clear and would seem to provide RaMP with a potentially unfair advantage if the baselines are not allowed to do the same. This needs to be better explained in any case. Furthermore, I'm not entirely sure why RaMP's performance even changes over time in the online planning phase since according to the pseudocode in Appendix A, all the learning happens before this. This too should be clarified.

Minor Comments, Questions and Suggestions
=========================================
* Section 2.1: "we can enable very quick (even zero-shot) adaptation to the new reward functions encountered at test time", what does "zero-shot" mean in this context?
* Section 4.3: "jumping and running backward have drastically different objectives that are difficult to express as parametric 'goals'", could you elaborate on this? I don't understand how it is the case, especially how running backwards is any harder to parameterize this way than running forward.
* Section 4.3: "MBPO fails to match the performance of RaMP since higher dimensional observation and longer horizon increase the compounding error of model-based methods", this is unclear to me and seems like an unproven claim. As mentioned above how do we know that MBPO couldn't be improved by for example adding more model rollouts.

**Summary Of The Paper:**

This paper presents an approach to model-free transfer learning in RL called Random Features for Model-Free Planning (RaMP). RaMP works by learning fixed horizon open-loop value functions (predicting the total reward received if H specific actions are selected in sequence starting from a state) for each of k randomly generated reward functions. When a new problem is encountered the algorithm learns to predict the rewards as a linear combination of the k random rewards via least squares. The fixed horizon open-loop action values for the new problem are likewise estimated as a waited combination of the k-learned value functions using these least squares coefficients, leveraging the linearity of fixed horizon open-loop value functions. These value function estimates are then used to select an action in the new environment by random shooting, that is randomly generating N action sequences and selecting the best according to the value function approximation. RaMP is experimentally compared in simulated robotics domains to a number of other transfer learning approaches, including model-based MBPO.

**Summary Of The Review:**

The method presented in this paper is potentially interesting as a reasonable baseline for transfer learning in certain problems. Further work in this direction has the potential to produce something even more interesting and broadly applicable. However, I believe that in the current version of the paper there are serious issues with the presented theory. In addition, I feel the generality of the method is overstated as the potentially detrimental effects of stochasticity are not discussed. It's not clear that the empirical conclusions are very robust as nothing is said about how hyperparameters are chosen for different algorithms and certain experimental details are unclear.

---

### Official Review · Reviewer_puvd · 2022-10-25

**Confidence:** 4
**Correctness:** 3
**Technical Novelty And Significance:** 3
**Empirical Novelty And Significance:** 2
**Recommendation:** 5

**Clarity, Quality, Novelty And Reproducibility:**

The paper is clearly written.
The work is quite related to successor features but it still different enough.

**Strength And Weaknesses:**

Strengths
- Authors propose a new way to exploit off-line trajectories without the need of reward labels.

Weaknesses
- The finite horizon fixes the dependency of the Q function on the policy. However, it makes longer time horizon tasks intractable. This is a major weakness of the proposed approach.
- Trajectories are generated with random shooting. It's unclear how this would scale to continuous action spaces or longer horizon tasks.
- The main results (Figure 3) do not show a clear picture of the results.
  - How good are the reported results? That is, what would the score of a standard algorithm like PPO be in these cases (given enough training data). Obviously, this is not a 1 to 1 comparison, but it is important to know if the reported scores correspond to almost optimal policies or if the agents just very quickly learn a simple policy that is far from good.
  - What happens if training goes on for longer? MBPO is quickly improving in 5 of the 8 plots and it seems that if the plots were just a bit longer, the results would be much more positive for MBPO.
  - RaMP seems to converge quickly and not improve anymore. This seems reasonable given that only the linear combination of Q-basis is learned. Can the learnt policy continue improving somehow? If the proposed approach quickly learns a degenerate policy but it can't improve afterwards, it's unclear the proposed approach would have any applications (unless the point RaMP converges to is already very good).
- It's unclear that any reasonable number of random features will make the approach work in high dimensional spaces. The only experiments in this setup are in the Appendix and are not very convincing. Why do you not make a comparison to MBPO, SF or other offline RL approaches in this setup?

**Summary Of The Paper:**

The authors propose to learn the discounted sum of random features over finite horizons using offline trajectories. By using finite-horizons, the authors break the dependence of the Q function on $\pi$. Once this pre-training phase is done, the agent must learn to linearly combine these discounted sums to learn the new reward function. With this approach, the agent can do model-predictive control to act on the environment.


**Summary Of The Review:**

The authors present an interesting idea. However, the paper has major gaps and I can't recommend acceptance.

---

### Official Review · Reviewer_5Avm · 2022-10-25

**Confidence:** 3
**Correctness:** 4
**Technical Novelty And Significance:** 3
**Empirical Novelty And Significance:** 2
**Recommendation:** 8

**Clarity, Quality, Novelty And Reproducibility:**

- The paper is well-structured and clear, and the overall quality of the paper is good.
- The proposed approach is meaningfully novel, backed up with some theoretical background, and empirically well-evaluated.

**Strength And Weaknesses:**

Strengths

- Despite its similarity to successor features, the authors successfully present how the proposed method is different and what its advantages are compared to successor features.
- Thm.3.1, wihch incorporates multiple hyperparameters, provides the theoretical background and justification to the proposed method.
- The overall presentation (especially Fig.1) and writing is good and easy to follow.

Weaknesses

- It would be informative to see how each method that appears in the comparison does well for different areas of goals, because the manifold of the used reward functions could affect their performance.
- Analyzing the space complexity of the method, especially in comparison with prior methods (mainly successor features) would be useful.

**Summary Of The Paper:**

The authors propose to model long-term behaviors for random cumulants, or Q-basis functions, in an open-loop manner. At test time, they can solve a ridge regression problem to fit the coefficient vector for a new task given some environment samples. They present the MPC-based planning, which can be used to gather samples. The authors also suggest the theoretical background on the relationship between the approximation of Q-functions and the used number of features and the horizon and the number of episodes. They show the experimental results in multiple manipulation and locomotion environments.

**Summary Of The Review:**

I find the proposed work reasonably novel, and the presented theoretical results are helpful for the community. They provide meaningful empirical results to show the effectiveness of the proposed approach. The writing and presentation is good.

---

### Decision · Program_Chairs · 2023-01-20

**Decision:**

Reject

**Justification For Why Not Higher Score:**

Although this paper has an average score of 5.25, I still don't consider it as a borderline paper and recommend a rejection for two reasons. First, majority of the reviewers (3 out of 4) have negative ratings. Second, the applicability of the proposed method is limited.

**Justification For Why Not Lower Score:**

N/A

**Metareview: Summary, Strengths And Weaknesses:**

This paper combines the merits of model-based and model-free reinforcement learning and proposes to learn reward-agnostic models using offline trajectories which thereby enables the agent to transfer across tasks with new reward functions quickly.

Strengths:
1. The proposed method is novel and interesting.
2. This paper provides theoretical justification to support the proposed method.
3. This paper is well-written and easy to follow.

Weaknesses:
1. The major concern is the applicability of the proposed method. Overall, the proposed method does not lead to improved performance compared to the previous methods. Although the proposed method can indeed result in better efficiency, the potential application scenarios are not clear and its importance is not well discussed.
2. Some of the experiments are not convincing enough, such as comparison in longer horizon tasks and analysis of reasonable number of random features.


**Summary Of Ac-Reviewer Meeting:**

N/A